# Robot Bionic Vision Technologies: A Review

Hongxin Zhang [1] and Suan Lee [2],*

1    Jewxon Intelligent Technology Co., Ltd., No. 293, Gongren East Road, Jiaojiang District,
     Taizhou 318000, China
2    School of Computer Science, Semyung University, 65 Semyung-ro,
     Jecheon-si 27136, Chungcheongbuk-do, Korea
*    Correspondence: suanlee@semyung.ac.kr; Tel.: +82-43-649-1273

**Abstract:** The visual organ is important for animals to obtain information and understand the outside world; however, robots cannot do so without a visual system. At present, the vision technology of artificial intelligence has achieved automation and relatively simple intelligence; however, bionic vision equipment is not as dexterous and intelligent as the human eye. At present, robots can function as smartly as human beings; however, existing reviews of robot bionic vision are still limited. Robot bionic vision has been explored in view of humans and animals' visual principles and motion characteristics. In this study, the development history of robot bionic vision equipment and related technologies are discussed, the most representative binocular bionic and multi-eye compound eye bionic vision technologies are selected, and the existing technologies are reviewed; their prospects are discussed from the perspective of visual bionic control. This comprehensive study will serve as the most up-to-date source of information regarding developments in the field of robot bionic vision technology.

**Keywords:** artificial intelligence; robot bionic vision; optical devices; bionic eye; intelligent camera

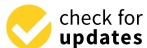



## 1. Introduction

Creatures and animals appeared on the earth as early as the Cambrian period. The visual organs of animals first appeared as photosensitive spots and evolved into more complex forms [1,2]. Among these, the most representative are the binocular stereoscopic visual organs of vertebrates and compound eye panoramic visual organs of insects [3–5]. Humans are the most advanced animals in nature. The human eye is an important organ for human beings to recognize and understand the external world. Humans' intellect is based on their sharp eyes and smart brains [6]. The human visual system is mainly composed of the eyeball, optic nerve, lateral geniculate body, and visual cortex of the brain [7]. The human eye possesses not only a sensing system but also a motor and analysis function [8]. Humans have always been curious about their eyes and brains [9], and Carlson [10] began studying the human brain in 1870. Berson proposed that "good camera equipment can provide stable images, and the movement of the retina and the vestibular system are the cameras of our eyes" [11–13]. For many years, most robot vision research has focused on the surface level of optical structures and vision algorithms in bionics. Currently, robot vision can only imitate one or a few physiological functions of the human eye and cannot gather, process, and evaluate images in complex and changing settings as the human eye does. With advancements in computer vision technology, many researchers have focused on the study of associated technologies [14–20]. Through iterative technology, they intend to achieve bionic vision, proceed to understand and model bionic brain processing, and finally, produce robots that are as flexible and intelligent as humans. This paper reviews the development of machine vision equipment and related technologies, selects the most representative binocular bionic vision technology and multi-eye bionic vision technology to display, and reviews the prospects of the existing technology from the perspective of

visual bionic control. This lays a foundation for further understanding the human visual system and researching humanoid visual electronic systems, Table 1 The history of related techniques' research is presented.

**Table 1.** Representative related research.

| Year | Survey Title | Reference | Focus |
|------|-------------|-----------|-------|
| 1859 | THE ORIGIN OF SPECIES | Darwin, C | Established the theory of biological evolution and studied the origin of everything [2]. |
| 1870 | The excitable cerebral cortex | Chad Carlson···. | Pioneered the study of the human cortex [10]. |
| 1963 | Variable Feedback Experiments Testing a Sampled Data Model for Eye Tracking Movements | Young, L R Stark, L. | Created models of machines that imitate human vision, saccades, etc. [21]. |
| 1968 | The Oculomotor Control System: A Review | DAVID A. ROBINSON. | The saccade, smooth movement, convergence, and control systems of eye movement were studied [22]. |
| 1970 | Oculomotor Unit Behavior in the Monkey | Robinson D A. | The relationship between the firing rate of motor neurons and eye position and movement was revealed by recording the oculomotor nerves of awake monkeys [23]. |
| 1986 | A model of the smooth pursuit eye movement system | D. A. Robinson, J. L. Gordon *, and S. E, Gordon. | Through research on macaques, a smooth-tracking-movement model of human eyes was created [24]. |
| 1987 | Visual motion processing and sensory-motor integration for the smooth pursuit of eye movements | Lisberger S G, Morris E J Tychsen | The proposed smooth-tracking model improves the visual tracking effect and overcomes the contradiction between high gain and large delay [25]. |
| 1989 | Dynamical neural network organization of the visual pursuit system | D.C. Deno, E.L. Keller, W.F, Crandall. | The dynamic neural network model was extended to a smooth-tracking system [26]. |
| 1998 | Neural adaptive predictor for visual tracking system | Lunghi, F Lazzari, S Magenes, G | An adaptive predictor was designed to simulate the prediction mechanism of the brain for visual tracking [27]. |
| 1992 | Adaptive feedback control models of the vestibulocerebellum and spinocerebellum | GOMI, H.; KAWATO, M. | An adaptive feedback control model was proposed, which is helpful for the vestibular eye reflex, the lobule and adaptive correction of eye movement [28]. |
| 1998 | Eye Finding via Face Detection for a Foveated, Active Vision System | Scassellati, B. | For the first time, robot eye interaction, image acquisition, and recognition functions were realized [29]. |
| 2002 | Quantitative Analysis of Catch-Up Saccades During Sustained Pursuit | De Brouwer S, Missal M Barnes G | A target-tracking experiment was carried out on macaques. Two conclusions were drawn: (1) there is a continuous overlap between saccade and smooth tracking; (2) the retinal sliding signal is shared by the two movements, which is different from the conclusion that the two traditional systems are completely separated [30]. |
| 2005 | Vestibular Perception and Action Employ Qualitatively Different Mechanisms. I. Frequency Response of VOR and Perceptual Responses During Translation and Tilt | Merfeld D M, Park S, Gianna-Poulin C, et al. | Established tVOR and rVOR models that simulate humans and studied the VOR and OKR vestibular-reflex eye-movement models through the ICub robot, which proved the important role of the cerebellum in image stabilization [31]. |

**Table 1.** *Cont.*

| Year | Survey Title | Reference | Focus |
|------|--------------|-----------|-------|
| 2006 | An Object Tracking System Based on Human Neural Pathways of Binocular Motor System | Xiaolin Zhang | A binocular motor system model based on the neural pathways of the human binocular motor system is proposed. Using this model, an active camera control system was constructed [32]. |
| 2006 | Design of a Humanoid Robot Eye: Models and Experiments | Cannata, G.; D'Andrea, M.; Maggiali, M. | By quantitatively comparing the performance of the robot's eyes with the physiological data of humans and primates during saccades, the hypothesis that the geometry of the eyes and their driving system (extraocular muscles) are closely related was verified [33]. |
| 2014 | Binocular Initial Location and Extrinsic Parameters Real-time Calculation for Bionic Eye System | Qingbin Wang, Wei Zou, Feng Zhang and De Xu | A simple binocular vision device was designed, using hand–eye calibration and an algorithm model to ensure the depth perception of the binocular vision system [34]. |
| 2017 | Design of Anthropomorphic Robot Bionic Eyes | Di Fan, Xiaopeng Chen, Taoran Zhang, Xu Chen, Guilin Liu··· | A new type of anthropomorphic robot with bionic eyes was proposed. A compact series-parallel eye mechanism with 3 degrees of freedom was designed to realize the eyeball movement without eccentricity [35]. |
| 2020 | Real-Time Robust Stereo Visual SLAM System Based on Bionic Eyes | Yanqing Liu, Dongchen Zhu, Xiaolin Zhang, | A real-time stereo vision SLAM system based on bionic eyes was designed that performed all movements of the human eye [36]. |
| 2021 | Panoramic Stereo Imaging of a Bionic Compound-Eye Based on Binocular Vision | Wang, X.; Li, D.; Zhang, G. | The optical optimization design scheme and algorithm for panoramic imaging based on binocular stereo vision were proposed, and a panoramic stereo real-time imaging system was developed [37]. |

* Source: Web of Science and Google Science.

## 2. Literature Review

Over the years, many academics have investigated the principles, structure, and characteristics of the animal and human visual systems, particularly the bionic vision system. For example, Carlson [10] mimicked both human and animal visual processing and eye-movement properties, creating a model that adequately mimicked human eye properties. Young [38–42] was a pioneer in the study of the human brain cortex in the twentieth century. Robinson [22,43–45] conducted an extensive study on human vestibular movements at MIT and developed a motion model for the human eye. Hubel [46,47] investigated the structure of the visual cortex through his work on eye-movement control and visual neural networks and made seminal contributions to the bionic study of human vision and eye movement that ultimately earned him the Nobel Prize. With the advancement of science and technology, several research institutes, such as MIT, Tokyo Institute of Technology, and the Chinese Academy of Sciences Bionic Machine Vision Laboratory have contributed relevant research. So far, robot eyes have been able to mimic some of the traits and functions of the human eye.

### 2.1. Human Visual System

Humans and other primates have distinct visual systems. How do you make a robot's eyes as flexible as a human's so that it can understand things quickly and efficiently? Through the study of the human and primate visual systems, we found that the human visual system has the following characteristics. Figure 1 depicts the human visual system's structure.

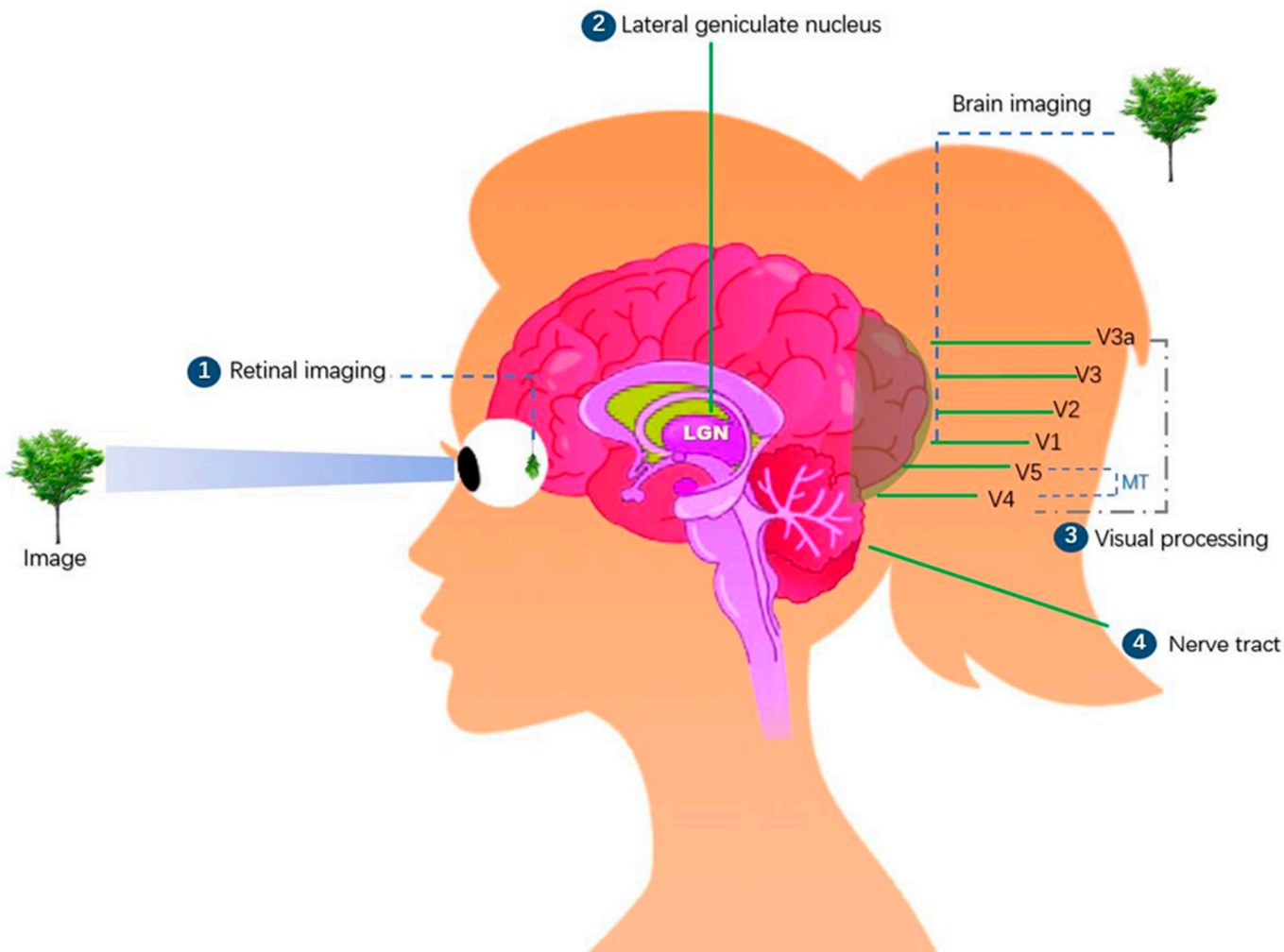

**Figure 1.** In the human visual system, "Retinal imaging" (①) is the first stage for processing an image when it enters the eye's visual field. The retina transmits pixels, shade, and color through the chiasma to the brain stem's "lateral geniculate nucleus" (LGN) (②). Visual information is further processed here, important visual information is extracted, useless information is discarded, and critical visual information is processed and transmitted to the primary "visual cortex" (③) V1 and then via V2 and V3 to V4, V5 (MT area), and higher brain areas [48,49]. The visual flow is then processed layer by layer and transmitted to more areas on the surface of the brain, where the brain identifies the object in the eye's view [50]. The "nerve tract" (④) runs through the brain and spinal cord throughout the body, and its main function is to control body movement [51].

In nature, the human eye is not the strongest. Animals such as eagles and octopi have eyes more developed than those of humans. What makes humans the most powerful animals in nature? This is the fact that the human brain is highly powerful [52–54]. To understand the deeper principles of the human visual processing system, let us examine the principles of human visual pathways. As shown in Figure 2, the human brain is divided into the left and right brains. The left brain is responsible for the visual field information of the right eye, while the right brain is responsible for the visual field of the left eye. Light waves are projected onto the retina at the bottom of the eyeball through the pupil, lens, and vitreous of the human eye, thereby forming a light path for vision. The optic nerve carries information from both the eyes to the suprachiasmatic nucleus. It then passes to the lateral geniculate nucleus (LGN), where visual information is broken down, twisted, assembled, packaged, and transmitted via optic radiation to the primary visual cortex V1, which detects information regarding the image's orientation, color, the direction of

motion, and position in the field of vision. The image information is then projected onto the ventral stream, also known as the "what" pathway, and the dorsal stream, also known as the "where " pathway, which is responsible for object recognition. The dorsal pathway is responsible for processing information such as movement and spatial orientation [55–58].

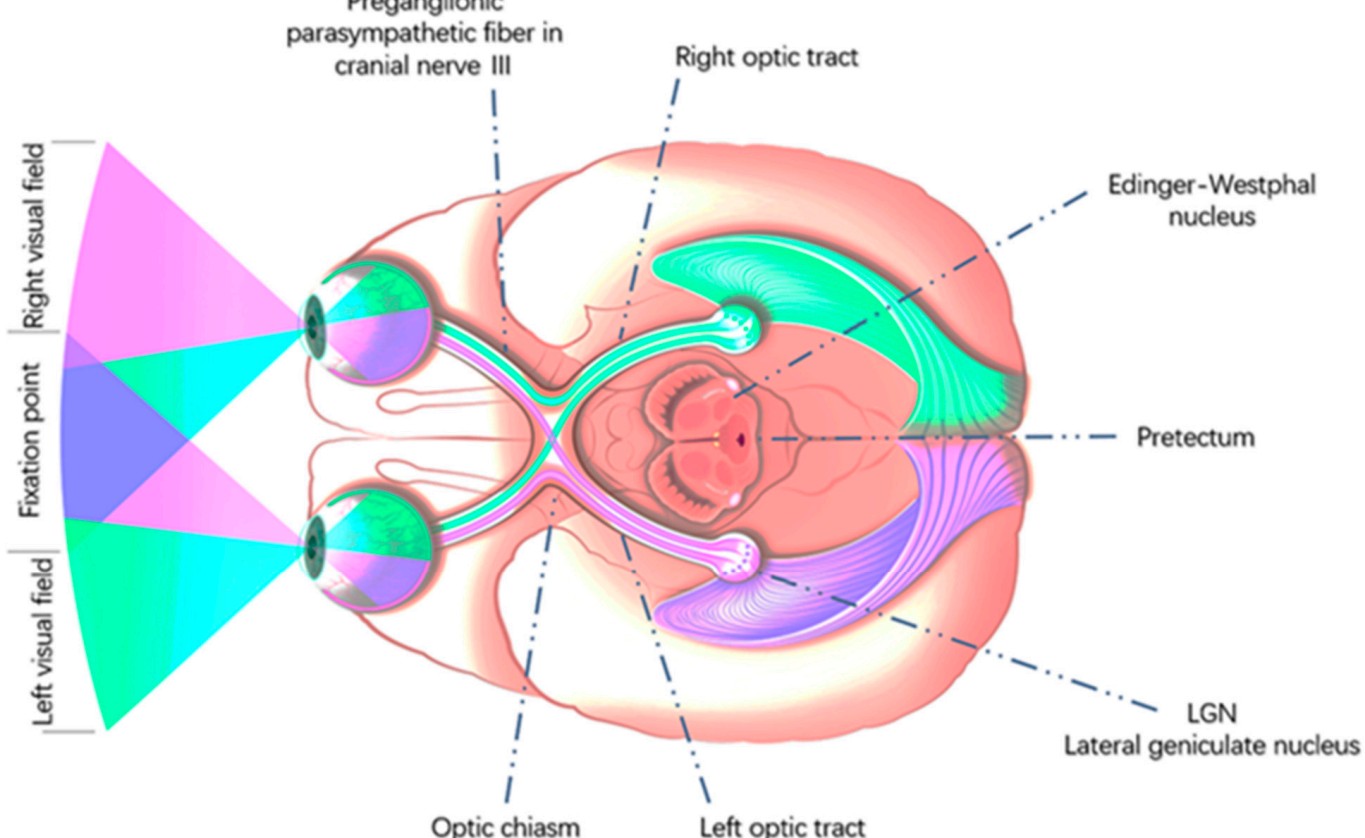

**Figure 2.** Human visual pathways. Copyright citation: This image is licensed by Eduards Normaals.

The superior colliculus controls the unconscious eye and head movements, automatically aiming the eye at the target of interest in the field of vision and making it shoot into the fovea of the retina of the human eye. This area contains approximately 10 percent of the neurons in the human brain and is the primary area for visual projection in mammals and vertebrates. The anterior tectum controls iris activity to regulate pupil size [59]. As shown in Figure 2, light information first enters the Edinger–Westphal nucleus through the anterior tectum. The Edinger–Westphal nucleus has the following functions: (i) controlling the ciliary muscle to regulate the shape of the lens; (ii) controlling the ciliary ganglion; and (iii) regulating pupil size via cranial nerve III [60].

Borra conducted anatomical experiments and studies on visual control in the rhesus-monkey brain. Stimuli and scans of the visual cortex of awake rhesus-monkey brains revealed that humans and nonhuman primates have similar visual control systems. However, the human brain is more complicated. Therefore, the similarity between the brain visual systems of rhesus monkeys and humans provides an important basis for humans to understand the principles of their eye movements and conduct experimental research [61,62]. The visual cortex regions of the human and macaque brains are depicted in Figure 3.

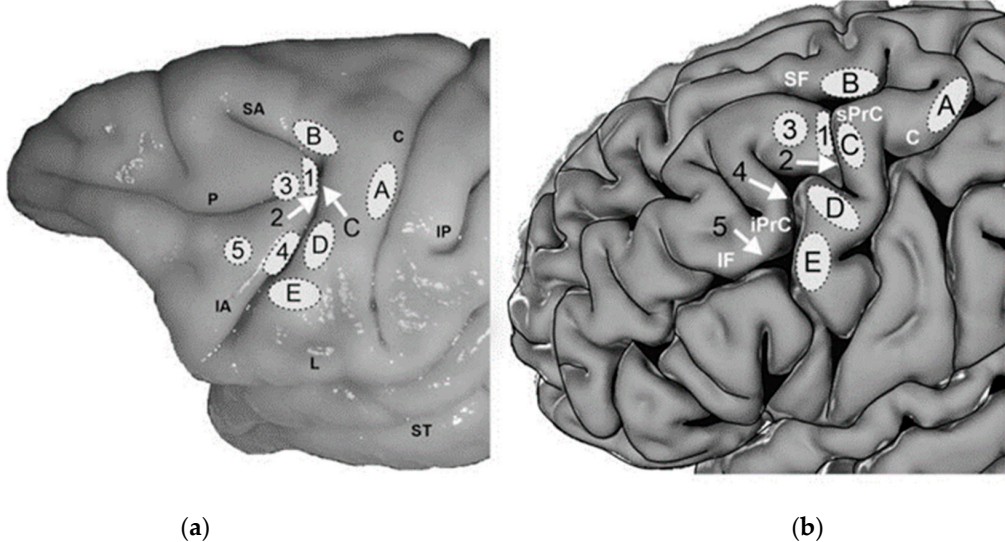

(**a**)                                                         (**b**)

**Figure 3.** A region at the junction of the prefrontal and premotor cortex in (**a**) rhesus monkeys and primates, (**b**) human brain. Known as the anterior field of vision (FEF), it is primarily involved in controlling eye-movement behavior and spatial attention. Humans have at least two types of forwarding fields of vision. After comparing the letters in the visual nerve regions of the human brain and monkey brain, it was found that the monkey visual cortex corresponds to the human visual cortex; and the principles of spatial attention, dynamic vision control, and attention orientation control are very similar [63]. The letters A, B, C, D, and E in the figure represent the visual cortex regions of the human and monkey brains. The image has been licensed by ELSEVIER.

Most mammals have two eyes that allow them to perform complex movements. Each eye is coordinated by six extraocular muscles according to [64]: the inner, outer, upper, and lower rectus muscles and the upper and lower oblique muscles. These muscles coordinate movements, allowing the eye to move freely in all directions and to change the line of sight as needed. The function of the internal and external rectus muscles is to turn the eyes inward and outward, respectively. Because the superior and inferior rectus muscles have an angle of 23° with the optic axis and the superior and inferior oblique muscles have an angle of 51° with the optic axis, they play a secondary role in addition to their main roles. The main function of the superior rectus muscle is upward rotation, whereas its secondary function is internal rotation.

The main function of the inferior rectus muscle is downward rotation, and its secondary functions are internal and external rotation. The main function of the superior oblique is internal rotation, and the secondary functions are external and downward rotation. The main function of the inferior oblique is external rotation, and the secondary functions are external and upward rotation. In addition, the movement of the extraocular muscles is restricted to prevent the eyes from going beyond the range of motion, and mutually restricted extraocular muscles are called antagonistic muscles. Synergistic muscles are cooperative in one direction and antagonistic in another. Movement of the eyeball in all directions requires fine coordination of the external eye muscles, and no eye muscle activity is isolated. Regardless of the direction of the eye movement, 12 extraocular muscles are involved. When a certain extraocular muscle contracts, its synergistic, antagonistic, and partner muscles must act simultaneously and equally to ensure that the image is accurately projected onto the corresponding parts of the retinas of the two eyes [65,66].

The extraocular muscle is driven and controlled by the cranial nerve, which directs the muscle to contract or relax. Eye movements occur when one part of the extraocular muscle contracts and the other relaxes. The motion features of the human eyes mainly include conjugate movement, vergence or disjunctive movement, vestibulo-ocular reflex (VOR), and optokinetic reflex (OKR) [63]. Specifically, the human eye also exhibits the following

motion characteristics [67–73]. The saccade, the most common feature, is a movement occurring rapidly and in parallel from one fixation point to another during which the human eye freely observes its surroundings, lasting between 10 and 80 ms. Mainly used by the human eye to actively adjust the direction of gaze, large saccades require head and neck movement. Smooth pursuit is the movement of the eye tracking a moving object, which is characterized by smooth tracking of the target. The angular velocity of the eye movement corresponds to the velocity of the moving object when the eye is observing a low-speed object. The adult eye can follow objects with 100% smoothness. When the target angular velocity is greater than 150°/s, the human eye cannot track the target. When the target object appears in the visual field of the human eye and begins to move at a high speed, the human eye tracks the target. The eyes have rhythmic horizontal nystagmus and maintain the same direction as the movement of the object. When the direction and speed of the human eye are consistent with those of the object, the human eye can clearly see the object. For example, one can sit in a fast-moving car and see the scenery outside the window. When the head moves, the eyes move in the opposite direction; thus, the image maintains steady reflective movement on the retina. For example, when the head moves to the right, the eyes move to the left, and vice versa. The primary function of OVR is motion stabilization, which is characterized by the coordination of head movements to obtain a more stable image. OVR has been used in camera anti-shake mechanisms. Conjugate motion is the simultaneous movement of the eyes in the same direction so that diplopia does not occur. Its characteristics can be divided into saccades and smooth tracking motions. When a target approaches, the visual axes of the two eyes intersect inward, presenting a convergent motion state, whereas when the target moves far away, the visual axis of the two eyes diverges outward, presenting a state of divergent motion. The hold time of the convergent motion is approximately 180 ms, and that of the divergent motion is approximately 200 ms. The process of keeping one's eyes on the target for a long time is called fixation. The eyeball quivers during the gaze, keeping the image refreshed and achieving a clearer visual effect. The eyes of most mammals vibrate slightly when focusing on stationary objects. The mean amplitude of a saccade is 0.2 ms (millisecond), which is half of the cone cell diameter. The vibration frequency reaches 80–100 Hz. Movement can produce light and light effects, and its purpose is to keep the visual center and visual cells in a state of excitement and maintain a high degree of visual sensitivity.

When we observe a scene through our eyes, our eyes do not perform a single motion feature but combine the various eye movement features mentioned above to achieve high-definition, wide-dynamic-range, high-speed, and flexible-target visual tracking. Making machines have the same advantages as humans and animals is the research goal of future robots.

### 2.2. Differences and Similarities between Human Vision and Bionic Vision

The human visual system operates on a network of interconnected cortical cells and organic neurons. In contrast, a bionic vision system runs on an electronic chip composed of transistors, mainly through a camera with CMOS and CCD sensors, image acquisition, and sending the image to a special image-processing system to obtain the form of the recorded target information and encode it according to pixel, brightness, color, and other information. The image-processing system performs various operations on these codes to extract the characteristics of the target. Through specific equipment, it simulates the visual ability of human beings to make corresponding decisions or execute on these decisions [74]. Figure 4 depicts the human vision processing and representative machine vision processing processes.

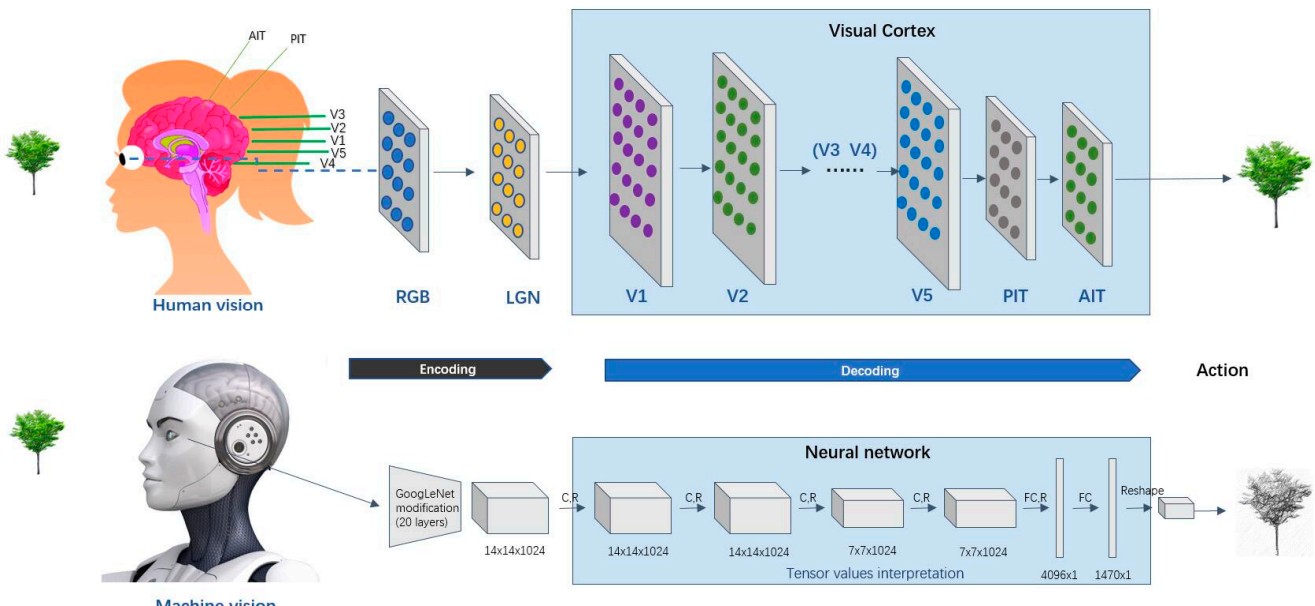

**Figure 4.** Human vision processing and Representative machine vision processing processes.

Human visual path: RGB light wave → cornea → pupil → lens (light refraction) → vitreous (supporting and fixing the eyeball) → retina (forming object image) → optic nerve (conducting visual information) → lateral geniculate body → optic chiasma → primary visual cortex V1 → other visual cortices V2, V3, V4, V5, etc. → the brain understands and executes → other nerves are innervated to perform corresponding actions.

Bionic vision path: RGB color image → image sensor (image coding) → image processing (image decoding) → image operator (image gray processing, image correction, image segmentation, image recognition, etc.) → output results and execution.

The information in the brain moves in multiple directions. Light signals move from the retina to the inferior temporal cortex, where they are transmitted to V1, V2, and other layers of the visual cortex. Simultaneously, each layer provides feedback on the previous layer. In each layer, neurons interact with each other and communicate information, and all interactions and connections essentially help the brain to fill in gaps in visual input and make inferences when information is incomplete.

The machine mainly converts the image into a digital electrical signal through the image acquisition device and transmits it to the image processor. There are plenty of image processors, and the most common one is GPU. With professional AI computing requirements, ASIC (application-specific integrated circuit) emerged. ASIC performs better than CPU, GPU, and other chips, with higher processing speed and lower power consumption. However, ASIC is very expensive to produce [75–77]. For example, Google has developed an ASIC-based TPU that is dedicated to accelerating the computing power of deep neural networks [78]. In addition, there are also NPU, DPU, etc. [79–81]. FPGA is a processor for developers to customize; usually, an image processor integrates machine vision perception and recognition algorithms. There are diverse algorithm models, such as CNN, RNN, R-CNN, Fast R-CNN, Mask R-CNN, YOLO, and SSD [82–84], which vary in terms of performance and processing speed. The most classical and widely used algorithm is the convolutional neural network (CNN) algorithm. Specifically, the convolutional layer first extracts the initial features, then the pooling layer extracts the main features, and finally, the fully connected layer summarizes all features to provide a classifier for prediction and recognition. The convolutional and pooling layers are sufficient to identify the complex image contents, such as faces and car license plates. Data in artificial neural networks often move in a one-way manner. CNN is a "feedforward network", where the information moves step by step from the input layer to the next layer and the output layer. This network

mainly makes use of the visual center of human beings, and its disadvantage is that it receives all continuous sequences. Utilizing CNN is equivalent to separating the front and the back connections. Therefore, the recursive neural network (RNN) that is dedicated to processing sequences emerges. In addition to entering the current information, each neuron has previously generated memory information that retained the sequence-dependent type. In the visual cortex, the information moves in multiple directions. The brain neurons are equipped with "complex temporal integration capabilities that are lacking in existing networks" [85]. Beyond CNN, Geoffrey Hinton et al. have been developing the "capsule network". Similar to the neuronal pathway, this new network is much better than the convolutional networks at identifying overlapping numbers [86].

In general, a high-performance image-processor platform is required for a high-precision image recognition algorithm. Although the great leaps in the fields of GPU rendering, hardware architecture, and computer graphics have enabled realistic scene and object rendering, these methods are demanding in terms of both time and computational resources [87]. As most robots are mobile, most bionic vision systems adopt a mobile terminal processor, and their performance cannot match a graphics server. For example, one of the features of the iCub robot is that it is equipped with a stereo camera rig in the head and a 320 × 240 resolution color camera in each eye. This setup consists of three actuated DoF in the neck of the robot to grant the roll, pitch, and yaw capabilities to the head, as well as three actuated DoF to model the oculomotor system of a human being (tilt, version, and vergence) [88]. Furthermore, F. Bottarel [89] made a comparison of the Mask R-CNN and Faster R-CNN algorithms using the bionic vision hardware of the iCub robot [90] and discovered that Mask R-CNN is superior to Faster R-CNN. Currently, many computer vision algorithm workers are constantly optimizing and compressing their algorithms to minimize the loss of accuracy and match the processing capacity of the end-load system. As shown in Table 2, the volume and accuracy of Mobilenet-Yolov3 surpass Mobilenet-SSD [91]. Moreover, Yuan, L., et al. designed the Florence algorithm, which performed well in ImageNet-1K zero-shot classification, with the top-1 accuracy of 83.74, the top-5 accuracy of 97.18, 62.4 mAP on COCO fine-tuning, 80.36 on VQA, and 87.8 on Kinetics-600 [92].

**Table 2.** Mobile-net and YOLO algorithm performance test data.

| Network | VOC mAP(0.5) | COCO mAP(0.5) | Resolution | Inference Time (NCNN/Kirin 990) | Inference Time (MNN arm82/Kirin 990) | FLOPS | Weight Size |
|---|---|---|---|---|---|---|---|
| MobileNetV2-YOLOv3-Lite(our) | 73.26 | 37.44 | 320 | 28.42 ms | 18 ms | 1.8 BFlops | 8.0 MB |
| MobileNetV2-YOLOv3-Nano(our) | 65.27 | 30.13 | 320 | 10.16 ms | 5 ms | 0.5 BFlops | 3.0 MB |
| MobileNetV2-YOLOv3 | 70.7 | & | 352 | 32.15 ms | & ms | 2.44 BFlops | 14.4 MB |
| MobileNet-SSD | 72.7 | & | 300 | 26.37 ms | & ms | & BFlops | 23.1 MB |
| YOLOv5s | & | 56.2 | 416 | 150.5 ms | & ms | 13.2 BFlops | 28.1 MB |
| YOLOv3-Tiny-Prn | & | 33.1 | 416 | 36.6 ms | & ms | 3.5 BFlops | 18.8 MB |
| YOLOv4-Tiny | & | 40.2 | 416 | 44.6 ms | & ms | 6.9 BFlops | 23.1 MB |
| YOLO-Nano | 69.1 | & | 416 | & ms | & ms | 4.57 BFlops | 4.0 MB |

Data sources: https://github.com/dog-qiuqiu/MobileNet-Yolo, accessed on 1 January 2021. Test platform: Mobile inference frameworks benchmark (4*ARM_CPU). The platform system is based on: NCNN and MNN. Conclusion: With a volume of 8.0 MB (15.1 MB less than SSD), Mobilenet-Yolov3 achieves 73.26% mAP (0.56% more than SSD).

Therefore, compared with machine vision, not only is the human vision system processing power great, but the human visual system also can dynamically modify the attention sensitivity in response to various targets. However, this flexibility is difficult to achieve in computer vision systems. Currently, computer vision systems are mostly intended for single purposes such as object classification, object placement, image region partitioning by

object, image content description, and fresh image production. There is still a gap between the high-level architecture of artificial neural networks and the functioning of the human visual brain [93].

Perception and recognition technology is the most critical technology for intelligent robots [94,95]. Perception technology based on visual sensors has been the most researched technology [96]. People use a variety of computer-vision-based methods to build a visual system with an initial "vision" function. However, the functions of the "eyes" of intelligent robots are still relatively low-level, especially in terms of binocular coordination, tracking of sudden changes or unknown moving targets, the contradiction between a large field of view and accurate tracking, and compensation for vision deviation caused by vibration. After millions of years of evolution, biological vision has developed and perfected the ability to adapt to internal and external environments. According to the different functional performances and applications of simulated biological vision, the research on visual bionics can be summarized into two aspects: one is research from the perspective of visual perception and cognition. The second is research from the perspective of eye movement and sight control. The former mainly studies the visual perception mechanism model, information feature extraction and processing mechanisms, and target search in complex scenes. The latter, based on the eye-movement control mechanism of humans and other primates, attempts to build the "eyes" of intelligent robots to achieve a variety of excellent biological eye functions [97–99].

### 2.3. Advantages and Disadvantages of Robot Bionic Vision

Compared to human vision, robot bionic vision has the following advantages [100–103]:

1.  High accuracy: Human vision is 64 gray-levels, and the resolution of small targets is low [104]. Machine vision can identify significantly more gray levels and resolve micron-scale targets. Human visual adaptability is considerably strong and can identify a target in a complex and changing environment. However, color identification is easily influenced by a person's psychology. Humans cannot identify color quantitatively, and their gray-level identification can also be considered poor, normally seeing only 64 grayscale levels. Their ability to resolve and identify tiny objects is weak.

2.  Fast: According to Potter [105], the human brain can process images seen by the human eye within 13 ms, which is converted into approximately 75 frames per second. The results extend beyond the 100 ms recognized in earlier studies [106]. Bionic vision can use a 1000 frames per second frame rate or higher to realize rapid recognition in high-speed image movement, which is impossible for human beings.

3.  High stability: Bionic vision detection equipment does not have fatigue problems, and there are no emotional fluctuations. Bionic vision is carefully executed according to the algorithm and requirements every time with high efficiency and stability. However, for large volumes of image detection or in the cases of high-speed or small-object detection, the human eye performs poorly. There is a relatively high rate of missed detections owing to fatigue or inability.

4.  Information integration and retention: The amount of information obtained by bionic vision is comprehensive and traceable, and the relevant information can be easily retained and integrated.

Robot bionic vision also has disadvantages such as a low level of intelligence, for it is unable to make subjective judgments, has poor adaptability, and has large initial investment cost. Furthermore, by incorporating non-visible light filling and photosensitive technology, bionic vision achieves night vision beyond the sensitivity range of the human eye. Bionic vision is not affected by severe environments, is not fatigued, can operate frequently, and has a low cost of continuous use. It simply requires an investment at the beginning to set up. Continued operation incurs only energy and maintenance costs.

### 2.4. The Development Process of Bionic Vision

If light from a source object is shone into a dark space through a small hole, an inverted image of the object can be projected onto a screen, such as the opposite wall. This phenomenon is called "pinhole imaging". In 1839, Daguerre of France invented the camera [107], and thus began the era of optical imaging technology. The stereo prism was invented in 1838. By 1880, Miguel, a handmade camera manufacturer in London, UK, had produced dry stereo cameras. At that time, people had early stereo vision devices, but these devices could only store images in films. In 1969, Willard S. Boyle and George E. Smith of Bell Laboratories invented a charge-coupled device (CCD), which is an important component of the camera [108]. A CCD is a photosensitive semiconductor. They can convert optical images into electrical signals. A tiny photosensitive material implanted in the CCD is called a pixel. The more pixels a CCD contains, the higher its picture resolution. However, the function of the CCD is similar to that of a film: it converts the optical signal into a charge signal. The current signal is amplified and converted into a digital signal to realize the acquisition, storage, transmission, processing, and reproduction of the image. In 1975, camera manufacturer Kodak invented the world's first digital camera using photosensitive CCD elements six years earlier than Sony [109]. At that time, the camera had 10,000 pixels, but the digital camera was still black and white. Surprisingly, in 2000, Sharp's J-SH04 mobile phone was equipped with a 110,000 pixel micro CCD image sensor. Thus began the age of phone cameras. People can now carry cameras in their pockets [110,111]. The development of vision equipment and bionic vision technology in Figure 5.

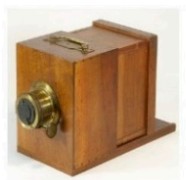 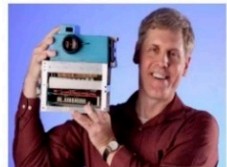 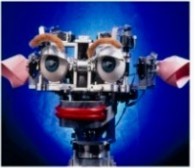 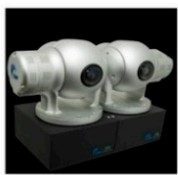 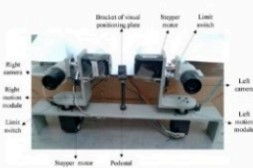 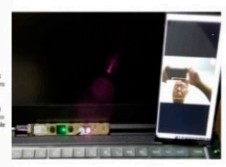

In 1839, Daguerre invented the world's first camera

In 1975, Kodak developed the world's first digital camera

Kismet Robot, was born at MIT Artificial Intelligence Laboratory, in 1998

In 2006, Zhang xiao lin has developed a bionic robotic eye that mimics the movements of the human eye

In 2014, Qing bin Wang, Wei Zou, et al., Bionic eye System developed at, Institute of Automation, Chinese Academy of Sciences.

In 2015, Intel Realsense 3D structured light vision system

**Early imaging system** — **Modern visual system**

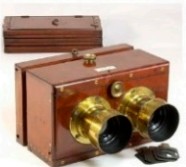 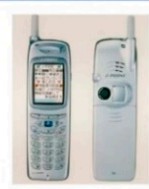 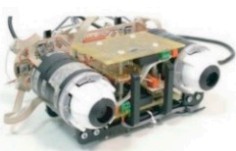 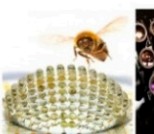 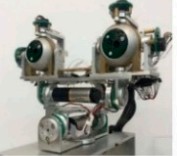 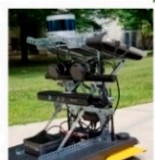

Dry plate camera, 1880, Meagher Company, London

In 2000, Sharp J-SH04, the first mobile phone to be equipped with a CCD micro camera

In 2006, University of Genova Giorgio Cannata and Mirko D'Andrea, The vision system of the binocular robot is developed

In 2013, The 160° Bionic Compound Eye Ultra Wide Angle vision system was developed by John Rogers of the University of Illinois and colleagues

In 2017, Di Fan, Xiao peng Chen, et al., Bionic eye System developed at Beijing Institute of Technology

MIT 2019 incorporates a fusion vision system for Lidar, structured light, and binocular stereo vision

**Figure 5.** The development of vision equipment and bionic vision technology.

Early image systems laid a solid foundation for modern vision systems and the future development of imaging techniques and processing. By consulting and searching through vast and relevant technical literature, it was found that the current industry mainly includes two technical schools: the bionic binocular vision system and the bionic compound eye vision system. The main research scholars are B. Scassellati, G. Canata, Z. Wei, Z. Xiaolin, and W. Xinhua.

According to WHO statistics, 285 million people worldwide are visually impaired. Of these, 39 million are classified as blind [112]. Scientists have been working to develop bionic eye devices that can restore vision. Arthur Lowery has studied this topic for several decades. His team successfully developed a bionic vision system called Gennaris [113].

Because the optic nerve of a blind person is damaged, signals from the retina cannot be transmitted to the "visual center" of the brain. The Gennaris bionic vision system can bypass damaged optic nerves. The system includes a "helmet" with a camera, wireless transmitter, processor, and $9 \times 9$ mm patch implanted into the brain. The camera on the helmet sends captured images to the visual processor, where they are processed into useful information. The processed data are wirelessly transmitted to the patch implanted into the brain and then converted into electrical pulses. Through microelectrode stimulation, images are generated in the visual cortex of the brain. Professor Fan Zhuo of the Hong Kong University of Science and Technology and his team successfully developed a bionic eye based on the hemispherical retina of a perovskite nanowire array. This bionic eye can achieve a high level of image resolution and is expected to be used in robots and scientific instruments [114,115].

In 1998, the Artificial Intelligence Laboratory at MIT created a vision system for the Kismet robot. However, at that time, robot vision could only achieve a considerably simple blinking function and image acquisition. Since then, significant progress has been made in binocular robot vision systems. In 2006, Cannata [33] from the University of Genoa designed a binocular robot eye by imitating the human eye and adding a simple eye-movement algorithm. At that time, the eyes of the robot could move rigidly up, down, left, and right. Subsequently, it was recognized by the Italian Institute of Technology (IIT) and applied to the iCub robot after continuous technical iterations [116]. In 2000, Honda developed the ASIMO robot [117]. Currently, the latest generation of ASIMO is equipped with binocular vision devices [118]. Thus far, most robots, including HRP-4C [119] and the robots being developed at Boston Dynamics, combine 3D LiDAR with a variety of recognition technologies to improve the reliability of robot vision [120]. Robots that have bionic vision in Figure 6.

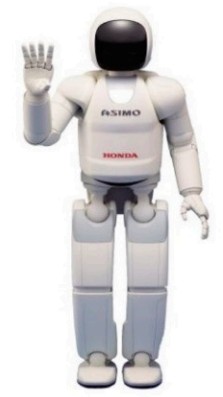
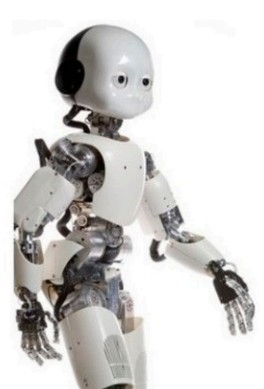
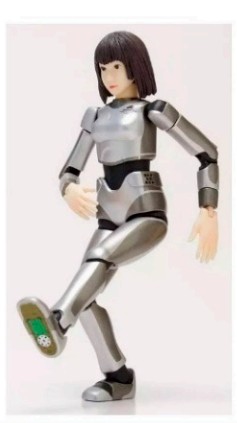
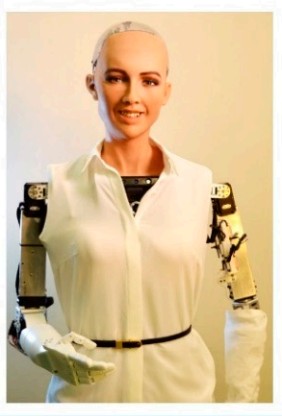
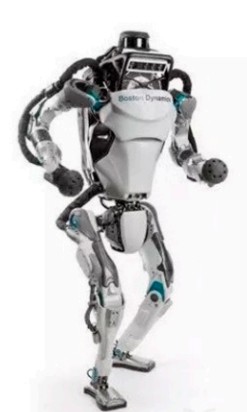

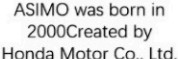

| ASIMO was born in 2000Created by Honda Motor Co., Ltd. | The iCub project was born in 2004 owing to the idea of Giorgio Metta, Giulio Sandini and David Vernon | HRP-4C in 2009From National Institute of Advanced Industrial Science and Technology | Sophia was born in 2015 Manufactured by Hanson Robotics Co., Ltd. | An upgraded version of Atlas humanoid robot in 2018, manufactured by Boston Dynamics |

**Figure 6.** Robots that have bionic vision. Copyright: Robot licensed by Maximalfocus US-L.

Machine vision processing technology is also being developed. As early as the 1980s, the Artificial Intelligence Laboratory at MIT began conducting relevant research. In the 1990s, bionic vision technology began to perform feature analysis and image extraction. In June 2000, after the first open-source version of OpenCV Alpha 3 was released [121], with the emergence of CNN, RNN, YOLO, ImageNet, and other technologies, bionic vision entered a stage of rapid development using artificial intelligence [122,123].

### 2.5. Common Bionic Vision Techniques

Bionic vision technology uses a camera equipped with a CCD or CMOS image sensor to obtain the depth information for an object or space. It is the most used technology

in intelligent devices such as robot vision systems and automatic vehicle driving. For example, NASA carried a binocular vision system on the Perseverance Mars Rover in 2020 [124]. In 2021, the Astronomy-1 Lunar Rover developed by China Aerospace also used a binocular terrain-navigation vision system [125]. At present, the mainstream technologies of binocular bionic vision include stereo vision, structured light, time of flight (TOF), etc. [96,126–140].

### 2.5.1. Binocular Stereo Vision

This technology was first introduced in the mid-1960s. It is a method for obtaining three-dimensional geometric information about an object by calculating the position deviation between the corresponding points on its images based on the parallax principle and using imaging equipment to obtain two images of the measured object from different positions. A binocular camera does not actively emit light; therefore, it is called a "passive depth camera". At present, it is used in the field of scientific research because it does not require a transmitter and receiver of structured light and TOF; thus, the hardware cost is low. Because it depends on natural light, it can be used indoors and outdoors, but strong and dark light can affect the quality of the image or the depth measurement of the object. This is a known disadvantage of the binocular camera. For example, for a solid color wall, because the binocular camera matches the image according to visual characteristics, a single background color causes a matching failure, and the visual matching algorithm is complex. The main representative products include Leap Motion and robotic eyes.

### 2.5.2. Structured Light

The basic principle of this technology is that the main hardware includes a projector and camera. The projector actively emits (therefore called active measurement) invisible infrared light onto the surface of the measured object, then captures pictures of the object through one or more cameras, collects structured light images, sends the data to a calculation unit, and calculates and obtains position and depth information through the triangulation principle to realize 3D reconstruction. Therefore, structured light is used to determine the structure of the object. There are many projection patterns, such as the sinusoidal fringe phase-shift method, binary-coded gray code, and phase-shift method + gray code. The advantages of this method include mature technology, low power consumption, low cost, active projection, suitability for weak lighting, high precision within a close range (within 1 m), and millimeter-level accuracy. A major disadvantage is poor long-distance accuracy. With the extension of the distance, the projection pattern becomes larger, and the accuracy worsens. It is not suitable for strong outdoor light, which can easily interfere with projection light. Representative products include Kinect, Primesense, and Intel Realsense.

### 2.5.3. TOF (Time of Flight)

This technology is a 3D imaging technology that has been applied to mobile phone cameras since 2018. Its principle is to emit continuous pulsed infrared light of a specific wavelength onto the target and then receive the optical signal returned by the target object via a specific sensor. The round-trip flight time or phase difference of the light is calculated to obtain depth information for the target object. The TOF lens is mainly composed of a light-emitting unit, optical lens, and image sensor. The TOF performs well in terms of close measurements and recognition. Its recognition distance can reach 0.4 m to 5 m. Unlike binocular cameras and structured light, which require algorithmic processing to output 3D data, TOF can directly output the 3D data of the target object. Although structured light technology is best suited for static scenes, TOF is more suitable for dynamic scenes. The main disadvantage is that its accuracy is poor, and it is difficult to achieve millimeter-level accuracy. Because of its requirements for time measurement equipment, it is not suitable for short-range (within 1 m) applications compared to structured light, which performs well for short- and high-precision applications. It cannot operate normally under strong light interference. Representative products that use this technology include TI-Opt and ST-Vl53.

It can be seen that Table 3 presents the following advantages and disadvantages: (1) monocular technology has a low cost and high speed, but its accuracy is limited, and it is unable to obtain depth information; (2) binocular ranging is the closest to human vision, without a passive light source, and the working efficiency is acceptable, but it is difficult to realize a high-precision algorithm, and it is easily disturbed by environmental factors; (3) a structured light scheme is more suitable for short-range measurements but has high power consumption; and (4) TOF and structured light have the same disadvantages. The miniaturization of the sensor has a significant impact on resolution. (5) Although LiDAR has high precision, even for long distances, its resolution is low. The machine vision measurement technologies have both advantages and disadvantages. Therefore, in recent years, many well-known robot manufacturers have adopted technology integration to improve the accuracy and practicality of robotic vision. For example, Atlas [141] of Boston Power adopts a vision combination of binocular and structured light + LiDAR.

**Table 3.** Performance comparison table of mainstream visual technology.

| Technology Category | Monocular Vision | Binocular Stereo Vision | Structured Light | TOF | Optical Laser Radar |
|---|---|---|---|---|---|
| Product pictures |  |  |  |  |  |
| Technology principle |  |  |  |  |  |
| Principle of work | Single camera | Dual camera | Camera and infrared projection patterns | Infrared reflection time difference | Time difference of laser pulse reflection |
| Response time | Fast | Medium | Slow | Medium | Medium |
| Weak light | Weak | Weak | Good | Good | Good |
| Bright light | Good | Good | Weak | Medium | Medium |
| Identification precision | Low | Low | Medium | Low | Medium |
| Resolving capability | High | High | Medium | Low | Low |
| Identification distance | Medium | Medium | Very short | Short | Far |
| Operation difficulty | Low | High | Medium | Low | High |
| Cost | Low | Medium | High | Medium | High |
| Power consumption | Low | Low | Medium | Low | High |
| Disadvantages | Low recognition accuracy, poor dark light | Dark light features are not obvious | High requirements for ambient light, short recognition distance | Low resolution, short recognition distance, limited by light intensity | Cloudy and rainy days, fog, and other weather interference have effects |
| Representative company | Cognex, Honda, Keyence | LeapMoTion, iit | Intel, Microsoft, PrimeSense | Intel, TI, ST, Pmd | Velodyne, Boston Dynamics |

Data sources: Official websites of related products and patent databases.

### 2.6. Robot Bionic Vision

Since the development of the first autonomous mobile robot, Shakey [142], by the Stanford Institute in the late 1960s, various robots have been developed. An important

device for robots to obtain external information is the robot vision system, which is an evolved version of bionic vision. It integrates bionic kinematics, visual algorithms, and adaptive deep-learning technology. The goal of robot vision is to realize a combination of the flexible movement of the human eye, understanding and memory of the human visual nervous system, and rapid image processing, which are mainly divided into two research directions:

### 2.6.1. Early Robot Vision System Model

In 1998, Breazeal created the Kismet robot at the Artificial Intelligence Laboratory of MIT. As shown in Figure 7, Kismet is an autonomous robot designed for social interactions with humans. The robot vision system consists of four-color CCD cameras installed on an active-vision head. Two wide field-of-view (FOV) cameras are mounted at the center and moved relative to the head. These are 0.25-inch CCD lipstick cameras with a 2.2 mm lens produced by Elmo. They are used to determine what the robot should pay attention to and for distance estimation. A camera is installed on the pupil of each eye. These are 0.5-inch CCD concave cameras with an 8 mm focal length lens for higher-resolution attention post-processing, such as eye detection. Kismet has three degrees of freedom to control the direction of sight and three degrees of freedom to control its neck. This motion is enabled using a Maxon DC servo motor. It is equipped with a high-resolution optical encoder to realize accurate position control. This enables robots to move and adjust the direction of their eyes like humans and participate in various human visual behaviors. Kismet can express emotions such as calm, interest, anger, happiness, depression, surprise, and nausea. The robot behaves like a human through facial expressions [143].

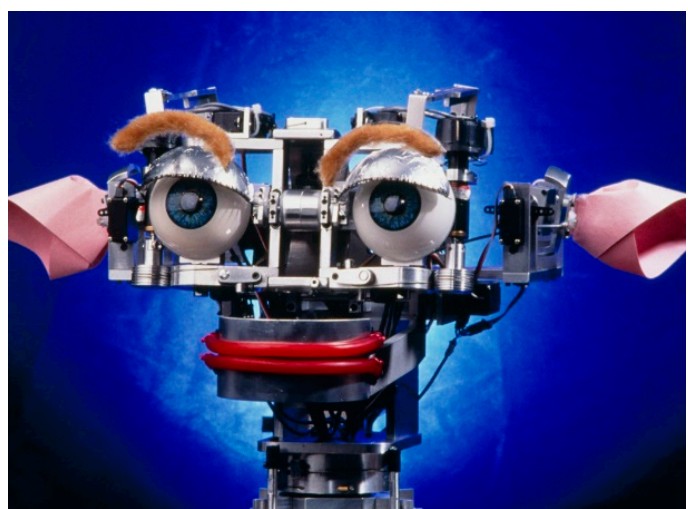 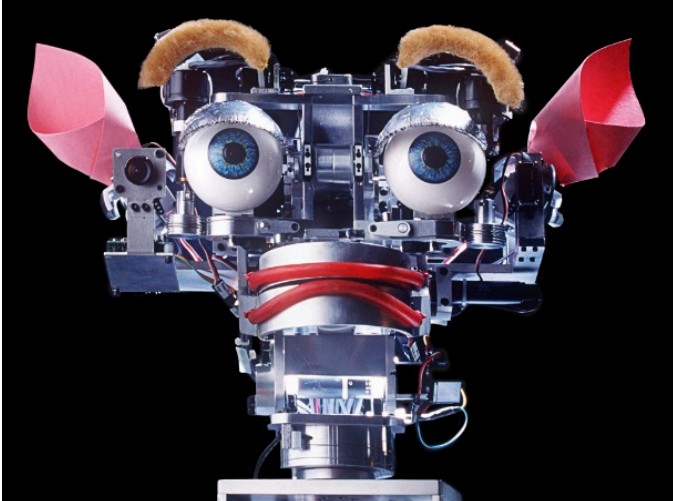

**Figure 7.** "Kismet" robot expression interactive vision system developed by MIT Artificial Intelligence Laboratory; Copyright citation: "MIT Museum" by Angela N. License CC (Attribution 2.0).

In 2006, Cannata [33] of the University of Genoa developed a robot bionic eye and proposed an eye-motion model and human eye muscle motion algorithm. As shown in Figure 8, It aimed to simulate the actual scanning and smooth tracking motions performed by the human eye. The system consists of a sphere (eyeball) driven by four independent tendons, which, in turn, are driven by four DC motors integrated into the system. The eyeball is equipped with a miniature CMOS color camera and is fixed by low-friction support that allows three degrees of freedom of rotation. Optical sensors provide feedback to control tendon tension during applications such as surgery. The control of the eyes was performed using an embedded controller connected to the host using the CAN bus. The proposed model was used for tendon-driven humanoid robot eyes [116]. Subsequently, this technology was applied to the neuron morphological control module of the real-time radiant eye movement of the iCub robot head.

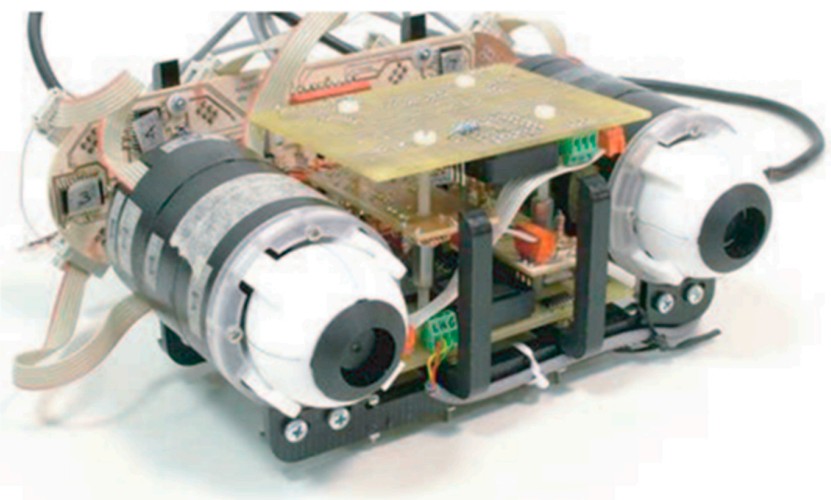

**Figure 8.** The eye model of a humanoid robot developed by Cannata and Maggiali. Copyright citation: This image is licensed by IEEE.

### 2.6.2. Robot Bionic Eye System

In 2006, Xiaolin Zhang realized the simulation function of human eye-motion characteristics using the algorithm of a binocular motion system to simulate the human eye-motion system according to the principle of the human eye-motion nervous system. As shown in Figure 9, Xiaolin Zhang's team has designed both equipment and an algorithm for bionic eye anthropomorphic bionic vision and realized the functions of six-degrees-of-freedom vision control and binocular stereo vision control. The device was developed based on the anatomical structure of the human eye and the mathematical model of a neural pathway, which uses a CCD camera as an eyeball and angle and acceleration sensors to simulate the function of the vestibular organs. The image processing and control module output a command to the servo motor (extraocular muscle) to control the eyeball and perform various movements. Figure 9 displays the various functions of the horizontal eye movement. The mechanical structure can cause the eye to move in three dimensions. However, the device can only realize the horizontal movement of the eyeball. The head and neck movements are also limited [144–151].

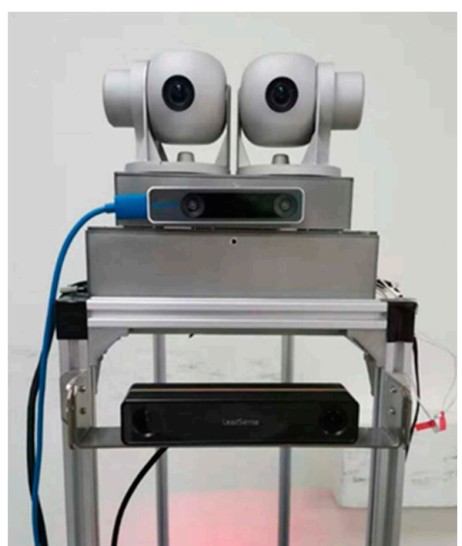 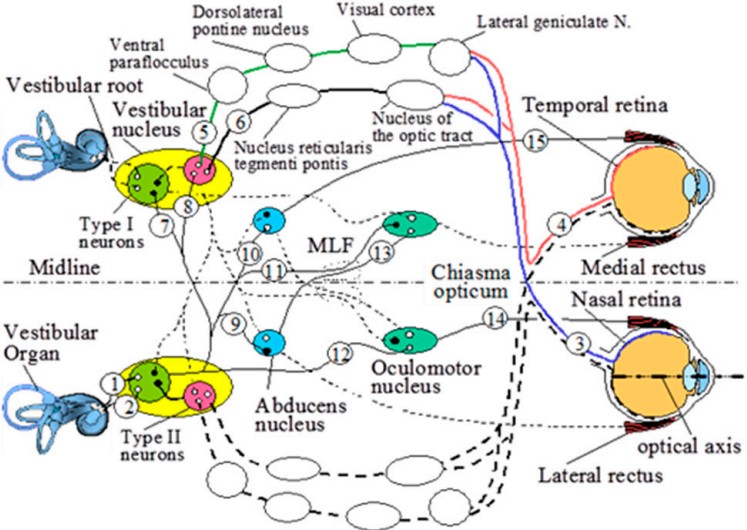

**Figure 9.** Bionic eye and algorithm model developed by Xiaolin Zhang's team. The visual pathway algorithm model that mimics the chiasma principle is depicted in ICONS: ①–⑮. Copyright citation: This image is licensed by IEEE.

To enable the robot to simultaneously have the humanoid environment perception ability of binocular stereo vision and monocular motion vision and overcome the defects of the narrow binocular field of view and low monocular depth-perception accuracy, Wei Zou's team designed a binocular bionic eye platform with four degrees of freedom based on the structural characteristics of human eyes and the control of scanning, As shown in Figure 10 smooth tracking, convergence, vestibular eye reflex, optokinetic reflex, and eye–head coordination. The equipment consists of two CCD cameras and four stepping motors. The initial positioning and parameter calibration of the platform was realized based on the visual alignment strategy and hand–eye calibration technology, respectively. The bionic eye platform uses a binocular stereo perception method based on the dynamic changes in external parameters and a monocular motion stereo perception method. The former performs three-dimensional perception through the image information obtained by two cameras in real-time and the relative pose information of cameras; the latter makes comprehensive use of multiple images and their corresponding gestures obtained using a single camera at multiple adjacent times for three-dimensional perception. Based on the consideration of the human eye-rotation angle error and image feature extraction error, a bionic human eye depth measurement error owing to the two active cameras was deduced. The factors affecting the depth measurement were obtained based on the calculation formula for the depth error. Based on an analysis of these influencing factors, an effective bionic eye measurement criterion was proposed to reduce measurement error: (i) observe the target as closely as possible, (ii) control the two active cameras at the same angular speed to ensure stability, (iii) effectively broaden the perception field of vision of traditional binocular vision, and (iv) ensure the accuracy of binocular and monocular motion perception [152–156].

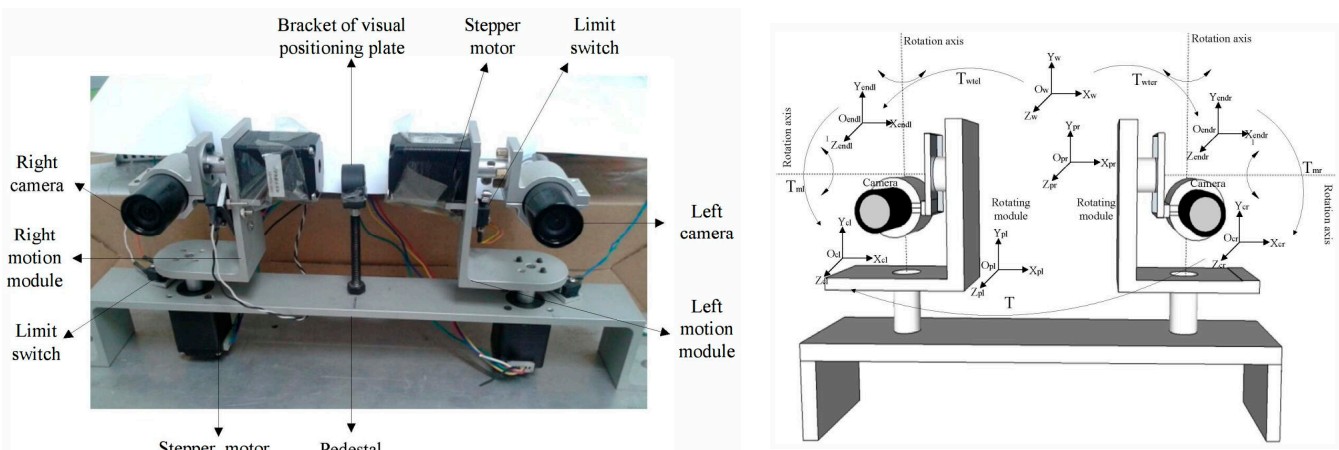

**Figure 10.** The bionic eye system mechanical structure and coordinate system. Copyright citation: This image is licensed by IEEE.

Most existing bionic eyes are continuous, without rotational degrees of freedom (DOF). However, the series eye mechanism is not sufficiently compact, and eye-rolling motion is the key to image stability. Therefore, Chen et al. designed a series-parallel eye mechanism with three degrees of freedom that can realize the non-eccentric spherical movement of the eye. As shown in Figure 11, The diameter of the eye is only 45 mm, and the total weight of the eye mechanism is 300 g. Each eye contains two cameras (long-focal-length and short-focal-length lenses) to simulate the perception of human eye features. They also designed a 3 DOF series neck mechanism to improve the imaging flexibility, ability, and stability of the robot's bionic eyes. The neck and each eye contain inertial sensors that can realize the bionic eye vestibular reflective hybrid image stabilization function of the robot. It is performed in two stages through the neck and eye mechanisms. Both robot bionic eyes contain the same number of DOFs and humanization, which is conducive to human–computer interaction. A real-time hybrid image stabilization method combining mechanical motion compensation

and electronic compensation was proposed. In the electronic image stabilization stage, the IMU generates the rotational motion of the three-dimensional camera and detects the two-dimensional image movement using the image-movement-detection algorithm. These motions are further filtered, and the image sequence is re-bent for a second time to remove the rotation jitter and translation jitter. The algorithm not only ensures real-time performance but also improves the fidelity of the inter-frame transformation [157].

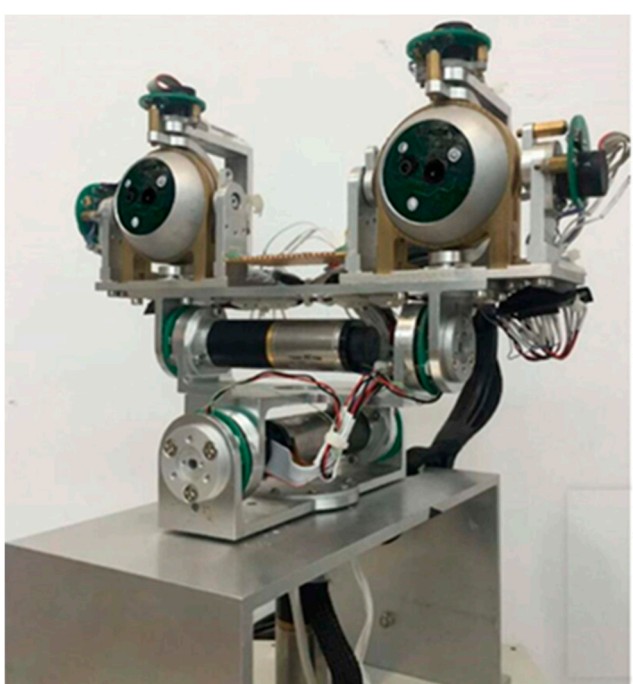 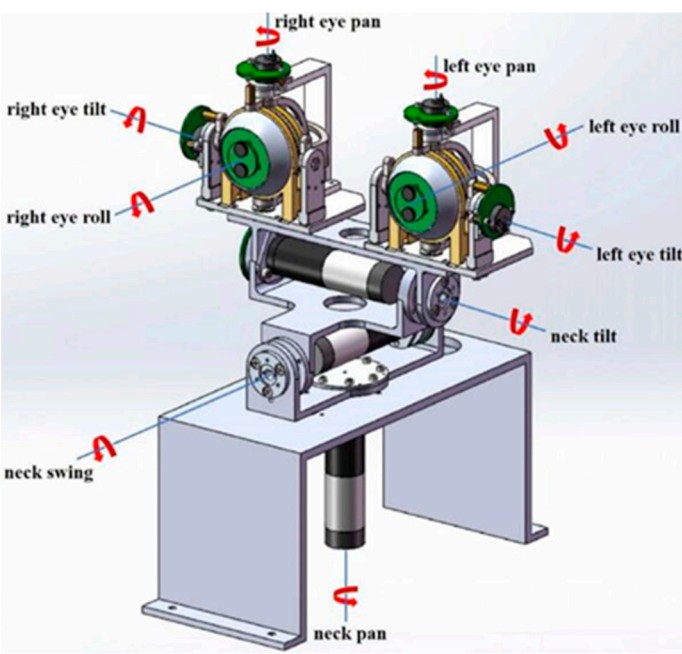

**Figure 11.** Bionic eye Vision Platform and Algorithm system. Copyright citation: This image is licensed by IEEE.

### 2.7. A Panoramic Imaging System Based on Binocular Stereo Vision

Compound eye technology has entered the field of mobile cameras. Currently, many new mobile phones are equipped with multiple cameras to achieve higher image quality through the combination of long-focus, medium-focus, and short-focus cameras. Then, Wang Xinhua and others at the School of Computer Science of Northeast Electric Power University and the State Key Laboratory of Applied Optics at the Changchun Institute of Optics, Precision Mechanics, and Physics, within the Chinese Academy of Sciences, refined the technology and designed a panoramic imaging system based on binocular stereo vision. This technology simulates the panoramic vision of the left and right eyes using an array camera and realizes the binocular stereo vision effect of a large field of view. The imaging principle of stereo vision is discussed based on a geometric model of binocular stereo vision. An optical optimization design scheme for panoramic imaging based on binocular stereo vision is proposed, and a spatial coordinate calibration platform for an ultra-high-precision panoramic camera based on the theodolite angle compensation function is constructed. The projection matrix of adjacent edges is obtained by solving the imaging principle of binocular stereo vision. Then, a real-time registration algorithm of multidetector stitched images and the Lucas Kanade optical-flow method based on image segmentation is proposed to realize stereo matching and depth-information estimation of panoramic images, and the estimation results are effectively analyzed. The experimental results show that the binocular panoramic stereo imaging system can satisfy the requirements for real-time and accurate image processing in virtual reality engineering applications.

### 2.8. Robot Bionic Vision Development Based on Human Vision

The human eye changes the shape of the lens through the contraction or relaxation of the ciliary muscle, thereby realizing zooming to obtain clear images of objects at different distances [54]. Referring to the zoom mechanism of the human eye, researchers have designed flexible zoom lenses with different driving modes and physical states [9]. Although the zooming process is different and the zooming effect is slightly different, most zooming processes achieve the zooming purpose by changing the curvature of the lens surface, and a few achieve the zooming purpose by changing the refractive index of the lens constituent material. The human eye changes the aperture of the pupil through the expansion and contraction of the smooth muscle on the iris (shrinks in bright light and expands in dark light), thereby controlling the amount of light entering the pupil to adapt to various lighting environment changes [8]. To enhance the imaging performance of the bionic visual optical system, in recent years, researchers have simulated the iris adjustment mechanism of the human eye, and have carried out many studies related to iris bionic technology. According to the differing focus of research, it can be roughly divided into two categories: the first category focuses on exploring the physiological model of pupil change, studies the relationship between human pupil diameter and brightness, and proposes a mathematical model of pupil diameter change based on adaptive adjustment of brightness. The other type focuses on the application of the iris driving mechanism in engineering. By simulating the telescopic motion mechanism of the smooth muscle on the iris, a freely adjustable optofluidic variable aperture optomechanical device is developed [59].

There are two different types of photoreceptors in the retina of the human eye: cone and rod cells. The two types of photoreceptor cells contribute differently to vision [65]. Cone cells sense strong light and color, have a strong resolution for object details and colors, and are mainly responsible for producing photopic vision in a brighter environment. Rod cells are sensitive to weak light and can only distinguish between light and dark but cannot distinguish the details and colors of objects. Rod cells are mainly responsible for producing scotopic vision in dim environments. These two types of photoreceptor cells are unevenly distributed in the retina [12]. There are approximately 7 million cone cells that are mainly distributed in the fovea area of the retina. The closer to the edge of the retina, the lower the density of cone cells, and the greater the density of rod cells. Based on the distribution and physiological characteristics of the above-mentioned retinal cone cells and rod cells, the details and color resolution of the human retina change with the change in spatial position; therefore, the retina can process information in a large field of view, high resolution, and real-time [13]. A balance between them can be achieved to meet the application requirements on different occasions. Inspired by this, in recent years, researchers at home and abroad have started to work on bionic research on retinal photoreceptors and imaging mechanisms. At present, research related to retinal bionic technology can be divided into two categories: the distribution characteristics of cone and rod cells and the retina's physiological characteristics [51,52].

According to the uneven distribution of retinal photoreceptor cells, only the target image in the fovea area has the characteristic of high resolution. However, the objects we observe are usually not static, and the scene changes. Therefore, to consider the need of a large field of view and high resolution, the entire eyeball needs to keep rotating, and the head and body movements also assist with changes in eyesight. Many physiological studies have shown that eye movement is a complex physiological process that includes the cooperation of brain nerve centers and multiple extraocular muscles to achieve the function of human eyesight shifting. To understand the eye-movement mechanism, simplify the human eye model, and make the structure and bionic kinematics of the human eye easy to realize, researchers proposed a hypothetical eye-movement center, eye-rotation axis, and Listing's law.

Some bionic vision optomechanical devices with anthropomorphic vision functions, intelligent image processing algorithms, and servo motion control models have been developed. However, after millions of years of continuous evolution, the human eye has

become a complete visual system that integrates image acquisition, information processing, and feedback, including the cornea, iris, lens, retina, optic neuron, and cerebral cortex visual center [135,136]. At present, the image acquisition, processing speed, and imaging performance of optomechanical devices, intelligent algorithms, and servo motion control models are far from those of the complex human visual system. This limits the application and development of intelligent robots in complex dynamic environments.

## 3. Challenges and Countermeasures for Robot Bionic Vision Development

### 3.1. Challenges That Restrict Robot Bionic Vision Development

The eye-movement control system is not only related to eye activity but also closely related to neural activity in the brain, cerebellum, and brain stem. From the medical perspective, this is an interdisciplinary and complex system. Therefore, the establishment of a machine eye-movement control system based on a biological vision mechanism is a research topic of interdisciplinary integration [158]. It is not only related to neurophysiology and medicine but also closely related to control science, bionics, biomedical engineering, and computer science. Although scholars at home and abroad have conducted many simulations and bionic studies on biological eye movement from different angles and for different purposes, there are still many problems to be further studied and discussed, mainly the following aspects:

- Although many scholars have conducted in-depth research on human vision, bionic vision still requires further research.
- At present, most robot vision systems use only left and right cameras to obtain and process target images. Left- and right-camera-driven systems have poor coordination, slow target tracking, and easy target loss. It is difficult to compensate for the line-of-sight deviation caused by image jitter in a complex motion environment. At present, researchers have constructed a head–eye motion control model of a bionic robot, which cannot fundamentally solve the problems faced by this version of this bionic machine [159].
- Existing bionic eye models usually only realize one or two movements of the human eye, and most are one-dimensional horizontal monocular or binocular movements. The separation mechanism of saccade motion and smooth tracking motion is widely used; however, its ability to realize multi-eye motion is poor.
- Binocular linkage control and head–eye coordination control in the line-of-sight transfer process are key technologies of bionic robot eyes. At present, research on binocular head–eye coordinated movement mostly remains at the level of biological neural control mechanisms and physiological models. Research on the construction of bionic binocular head-and-neck coordinated control is still in its infancy.
- The advanced mechanisms of human vision have not been fully explored, and some mechanisms remain controversial and undiscussed. Despite research and analysis from the perspectives of neurophysiology and anatomy, there has been no breakthrough in head–eye coordinated motion control technology. Therefore, it is necessary to establish new models, theories, and methods to make a machine's eyes more intelligent, dexterous, and realistic.

At present, most research on robot eyes is based on engineering methods. The left and right cameras are used to obtain the target images for processing. The left and right eyes and the head and neck lack a coordinated linkage mechanism. There are many technical obstacles, and the adoption of bionic technology is an important method to solve these problems. Research on bionic robotic eyes is multidisciplinary, involving neurophysiology, bionics, control science, and computer science. There are still many problems in the research on robot bionic vision that need to be explored further.

*3.2. Countermeasures That Can Promote Robot Bionic Vision Development*

3.2.1. Building a Complete Bionic Eye Model

Most foreign research is conducted by neurophysiologists. Through physiological experiments and data analysis, the neural mechanisms and pathological characteristics of eye movements are explored to provide technical support for clinical medicine. Most of these research results are limited to physiological models, and their engineering implementation and application have rarely been considered. Based on the research results of physiology, a combination of engineering bionics and control theory is used to transform the physiological model of vision control into an engineering technology model, thereby realizing the excellent performance of biological vision. This fundamentally solves the technical problems faced by machine vision and is an important direction for intelligent robot research. The bionic eye models in the current literature only simulate one or two movements of the human eye, and most of them are one-dimensional horizontal movements of one or both eyes. The separation mechanism of saccade and smooth tracking is generally used, and it is difficult to simultaneously achieve multiple eye movements. Further study of the internal correlation and neural mechanism between various eye movements, the establishment of a multi-degree-of-freedom nonlinear bionic binocular movement model, and realization of various eye movements such as saccades, smooth tracking, anisotropic movement, and reflex movement are required.

3.2.2. Achieving Robot Head–Eye Coordination through a Biological Neural Control Mechanism

The joint redundancy of the robot is required to effectively coordinate the head and binocular movements when performing large-scale gaze shifts. Most of the currently proposed head–eye coordination motion systems and learning algorithms are based on two-dimensional head–eye systems in the horizontal direction. Although some scholars have proposed 3D head–eye coordination motion control algorithms, they use binocular focusing first and then control the head rotation. The "time-sharing" and "segmented" execution methods of eye compensation do not realize coordinated control in the simultaneous turning of the head and eye to the target. Currently, 3D head–eye coordination remains a technical challenge. It is an important research objective to obtain inspiration from the coordinated movement of the biological head and eye, study the neural control mechanism of the coordinated combination of the head, eye, and body of primates to complete the sight shift, and design a control algorithm for the coordinated movement of the head and neck of the binocular.

3.2.3. Improving the Speed and Accuracy of Random-Moving-Target Tracking

One of the important functions of the "eye" of a robot is visual tracking. Currently, a visual servo feedback control method is often used. However, for sudden, fast-changing, and erratic targets, target loss and tracking failure often occur. The extraordinary ability of the human eye to track changing targets stems from the coordination and real-time switching between rapid saccade and the slow and smooth follow-up of the eye in its visual system. According to an in-depth study, the neurophysiological mechanism of automatic switching of human eye-movement mode to track the target quickly and accurately, the switching of saccade and smooth pursuit modes in visual tracking, is regarded as the most adaptive hybrid system. The optimal control problem is studied to solve the problem of the rapidity and accuracy of random-moving-target tracking. The optimal switching timing between the two modes and the prediction algorithm are the key technologies of this research.

3.2.4. Solving the Problem of Large-Scale Line-of-Sight Deviation Compensation

When the robot operates in a complex, unstructured environment, the vibration of its own body or the change in its posture causes a large line-of-sight deviation. Usually, image processing is used to adjust the servo mechanical head; however, the compensation range is small, the image stability is poor, and there is no way to solve the large line-

of-sight deviation. The human eye has strong self-adaptive and self-adjusting functions. When the head and body posture changes or the background changes dynamically, it can still clearly gaze at and track the target because of its vestibule–oculomotor reflex (VOR) and optokinetic reflex (OKR) functions. By studying the mechanism of human reflex eye movement, a feedback controller based on retinal sliding information and an adaptive VOR-OKR model can be established, which can actively compensate for the visual error caused by the change in the robot's body position and then solve the large-scale visual deviation compensation problem of the robot.

## 4. Conclusions

In today's smart era, humans desperately require more adaptable and intelligent machines to serve humanity. Bionic vision systems have a wide range of applications. However, robot vision remains in the early stages of bionics. Although exploration of the human visual system has laid a solid foundation for artificial-intelligence vision technology, at the current stage of human scientific exploration, machine realization of the human visual system remains in the V1 primary visual cortex. The specific mechanism of action of the advanced visual cortex requires further investigation. To date, robot vision has not been able to fully actualize all the capabilities of the human eye.

At present, many scientific research achievements remain in human eye physiological characteristics, eye-movement principles, and bionic eye prostheses, and research on the robot bionic eye is still in its primary stage. This paper summarized the research progress of artificial intelligence bionic vision technology at home and abroad by citing the relevant literature. By summarizing the visual principles and motion characteristics of humans and animals, robot vision was explored. This study meticulously examined the development history of bionic vision equipment and related technologies, discussed existing bionic vision technologies, and selected the most representative binocular, bionic, and multi-eye compound bionic vision technologies. Prospects of existing technologies from the standpoint of visual bionic control were discussed. Although this study showed that some research progress has been made in using biological vision mechanisms as a reference to study bionic vision control methods that mimic highly accurate and complex human vision, the current technology is only a preliminary attempt, and there are still many problems to be addressed.

Biological head–eye movement and visual control are not only related to the activities of the head and eyes but also related to the activities of the brain and central nervous system. The behavioral expression of brain control strategies is an interdisciplinary and highly complex research subject. Humans are familiar with and yet unfamiliar with their cognition. Perhaps soon, humans will fully understand the working principles of the entire visual cortex and uncover the mysteries of the brain. That will be the era of robots that can perform sophisticated human tasks.

**Author Contributions:** Conceptualization, H.Z.; methodology, H.Z.; validation, S.L.; formal analysis, H.Z.; investigation, H.Z.; resources, H.Z.; writing—original draft preparation, H.Z.; writing—review and editing, H.Z.; supervision, S.L.; project administration, S.L. All authors have read and agreed to the published version of the manuscript.

**Funding:** This paper was supported by the Semyung University Research Grant of 2021.

**Institutional Review Board Statement:** Not applicable.

**Informed Consent Statement:** Not applicable.

**Data Availability Statement:** The data presented in this study are available upon request from the corresponding author.

**Conflicts of Interest:** The authors declare no conflict of interest.

## Abbreviations

The following abbreviations are used in this manuscript:

| | |
|---|---|
| VOR | Vestibular ocular reflex |
| OKR | Objectives and key results |
| LGN | Lateral geniculate nucleus |
| RGB | Red–green–blue |
| FOV | Field of view |
| OpenCV | Open source computer vision library |
| CNN | Convolutional neural network |
| RNN | Recurrent neural network |
| NCNN | High-performance neural network inference computing framework |
| R-CNN | Region-Convolutional neural network |
| Fast R-CNN | Fast region-Convolutional neural network |
| Mask R-CNN | Mask region-Convolutional neural network |
| SSD | Single Shot multibox detector algorithm |
| YOLO | YOLO algorithm |
| MS | Millisecond |
| CCD | Charge-coupled device |
| COMS | Complementary metal-oxide semiconductor |
| TOF | Time of flight |
| DC | Direct current |
| CAN | Controller area network |
| DOF | Degree of freedom |
| IMU | Inertial measurement unit |
| CPU | Central processing unit |
| GPU | Graphics processing unit |
| DPU | Deep learning processing unit |
| NPU | Neural network processing unit |
| TPU | Tensor processing unit |
| FPGA | Field-programmable gate array |
| ASIC | Application-specific integrated circuit |

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
