# Peer review of "Robot Bionic Vision Technologies: A Review"

_applsci, doi:10.3390/app12167970_

Round 1

Reviewer 1 Report

In this paper, the research progress of artificial intelligence bionic vision technology at home and abroad is reviewed by searching the related literature on "bionic eye technology" on the Web of Science, Google Scholar, and IEEE Xplore, and citing peer research papers. The article has a good structure and is well written. So, the article is at an acceptable level and can be accepted after minor revisions:

1- Abstract: the authors have clearly mentioned the purpose, methods, results and conclusion. No comments

2- Introduction: the authors included the needs, importance of research topic, methods, and main contributions. Besides the paper structure. No comments.

3- Related works: the authors have clearly reviewed the and illustrated the most important parts in the state-of-the-art methods including lightweight network design to solve the trade-off between computation cost and overall accuracy. The only problem is that the image quality is not good.

4- Research characteristics and existing problems of bionic robot eye: This section has a lot of work to do. It is necessary to deal with open problems in more detail.

5- Conclusion: No Comments

6- References: the references provided applicable and sufficient.

The final decision for this paper is accepted with minor corrections. 

Author Response

Response to Reviewer 1 Comments

In this paper, the research progress of artificial intelligence bionic vision technology at home and abroad is reviewed by searching the related literature on "bionic eye technology" on the Web of Science, Google Scholar, and IEEE Xplore, and citing peer research papers. The article has a good structure and is well written. So, the article is at an acceptable level and can be accepted after minor revisions:

  • Thanks for your kind and constructive suggestions. We have carefully revised according to your suggestions.

1- Abstract: the authors have clearly mentioned the purpose, methods, results and conclusion. No comments

  • Thanks for your kind and constructive suggestions

2- Introduction: the authors included the needs, importance of research topic, methods, and main contributions. Besides the paper structure. No comments.

  • Thanks for your kind and constructive suggestions

3- Related works: the authors have clearly reviewed the and illustrated the most important parts in the state-of-the-art methods including lightweight network design to solve the trade-off between computation cost and overall accuracy. The only problem is that the image quality is not good.

  • Thanks for your kind reminding, we have exchanged the previous images with high resolution ones.

4- Research characteristics and existing problems of bionic robot eye: This section has a lot of work to do. It is necessary to deal with open problems in more detail.

  • Thanks for your kind reminding, we have reorganized this section. See line 638-744 in yellow.
  • Conclusion: No Comments
  • Thanks for your reviewing.

6- References: the references provided applicable and sufficient.

The final decision for this paper is accepted with minor corrections.

  • Thanks for your suggestions.

Reviewer 2 Report

  1. Authors need to introduce the difference and connection between computer vision, bionic vision, robot vision and machine vision. And Authors also need to make sure that the terminology used in a paragraph or section needs to be consistent.
  2. The paper needs to be polished by a native English speaker.
  3. The following content in the abstract is not helpful to the theme and is recommended to be deleted. "How can future robots have brains and eyes as smart as human beings? The existing reviews of bionic vision are one-sided and the discussion on technology is outdated. In this paper, the research progress of artificial intelligence bionic vision technology at home and abroad is reviewed by searching the related literature on "bionic eye technology" on the Web of Science, Google Scholar, and IEEE Xplore, and citing peer research papers."
  4. The structure of the article is unbalanced. For example, "section 2 related work" is a lump, which includes multiple sub-topics, but section 3 is not enough to be a separate section. I suggest adjusting the article's structure, such as splitting section 2.
  5. 7.1, 2.7.2, 2.7.3 and 2.7.5 should be a title like 2.7.4 /2.7.6, not a sentence
  6. Line 258, "The visual system can dynamically…" should be "The human visual system …"
  7. Section 2.4 only mentions the advantages of machine vision relative to human vision, but does not mention its disadvantages relative to human vision. In Line 258 – line 263, there is a mention of the current inadequacy of machine vision, so it can be expanded and explained in section 2.4
  8. Section 2.5 needs to be further expanded. Line 320 mentions "bionic binocular vision system" and "bionic compound eye vision system" but does not explain or compare them. I noticed that sections 2.7.1 and 2.7.2 are the introductions of them. Should it be a good idea to move 2.7.1 and 2.7.2 to this section? Also, the introduction to machine-vision processing technology is sketchy and needs to be expanded.
  9. Items in Table 2 should be compared using quantitative data as much as possible, such as Response Time, Identification distance etc.
  10. The paper focuses on the research related to the motion control of the robotic vision system but lacks an introduction to the back-end artificial intelligence algorithm for image processing.

Author Response

Response to Reviewer 2 Comments

Authors need to introduce the difference and connection between computer vision, bionic vision, robot vision and machine vision. And Authors also need to make sure that the terminology used in a paragraph or section needs to be consistent.

  • Thanks for your kind reminding and suggestions, we have carefully checked and unified the use of terminology.

The paper needs to be polished by a native English speaker.

  • Thanks for your kind reminding. Our paper has been polished by a native English speaker.

The following content in the abstract is not helpful to the theme and is recommended to be deleted. "How can future robots have brains and eyes as smart as human beings? The existing reviews of bionic vision are one-sided and the discussion on technology is outdated. In this paper, the research progress of artificial intelligence bionic vision technology at home and abroad is reviewed by searching the related literature on "bionic eye technology" on the Web of Science, Google Scholar, and IEEE Xplore, and citing peer research papers."

  • Thanks for your kind reminding. We have deleted this content according to your suggestions.

The structure of the article is unbalanced. For example, "section 2 related work" is a lump, which includes multiple sub-topics, but section 3 is not enough to be a separate section. I suggest adjusting the article's structure, such as splitting section 2.7.1, 2.7.2, 2.7.3 and 2.7.5 should be a title like 2.7.4 /2.7.6, not a sentence

  • Thanks for your kind and constructive suggestions. We have reorganized the structure of this paper. It can been see in section 2 and section 3.

Line 258, "The visual system can dynamically…" should be "The human visual system …"

  • Thanks for your kind reminding, we have changed this description according to your suggestion. See line 235-236 in yellow.

Section 2.4 only mentions the advantages of machine vision relative to human vision, but does not mention its disadvantages relative to human vision. In Line 258 – line 263, there is a mention of the current inadequacy of machine vision, so it can be expanded and explained in section 2.4

  • Thanks for your kind reminding. We have revised this issue according to your suggestions. See line 260-289 in section 2.3.

Section 2.5 needs to be further expanded. Line 320 mentions "bionic binocular vision system" and "bionic compound eye vision system" but does not explain or compare them. I noticed that sections 2.7.1 and 2.7.2 are the introductions of them. Should it be a good idea to move 2.7.1 and 2.7.2 to this section? Also, the introduction to machine-vision processing technology is sketchy and needs to be expanded.

  • Thanks for your kind reminding and constructive suggestions. We have revised according to your suggestions. See line 314-358 in yellow.

Items in Table 2 should be compared using quantitative data as much as possible, such as Response Time, Identification distance etc.

  • Thanks for your kind reminding. This quantitative data can not be reached in the open website.

The paper focuses on the research related to the motion control of the robotic vision system but lacks an introduction to the back-end artificial intelligence algorithm for image processing.

  • Thanks for your kind reminding. We have included the needs, importance of our research topic, methods, and main contributions in the introduction part.

Reviewer 3 Report

The authors proposed a review paper on robot bionic vision technologies. Overall, they aimed to summarize the developmental history of machine vision equipment and related technologies, mentioning the most representative binocular bionic and multi-eye compound eye bionic vision technologies, and review existing technologies and their prospects from the perspective of visual bionic control.

However, the manuscript is not ready for publication at least until several major short-comings are addressed. The actual contents have the following issues:

-          Discussion of hardware-software developmental aspects in robot bionic vision appear to have been significantly omitted. The authors have made no discussion on how biological optic research may have influenced or is relevant with hardware-software designs.

o   The authors have focused too much on describing past research works of biological optic functions, parts and their technicalities. They must make appropriate connections to actual design choices, technical reasons, motivations for robot bionic vision technologies that mimic biological functions.

o   The authors have not provided sufficient proof or clarity to support the correspondences between Human Visual paths and Machine Visual paths. Besides, Neural-Network functions are not the only approach to model neural functionalities of biological brains. Please make sure not only to generalize the concepts, but also to discuss the correspondences.

o   The authors have not elaborated the software aspects of Machine Visual paths with respect to actual biology. Please clarify and compare between current findings in computer vision and actual biological functions.

o   The authors’ statement is deniable in cases of noisy images or low resolution, major advances have only made better (but imperfect) predictions even with attention mechanism: “The convolution and pooling layers are sufficient to recognize all types of complex image content, such as human faces and auto-mobile license plates.”

-          The paper presents very few discussions and analyses.

o   According to the abstract, Section 2.7 is arguably a main part. However, it lacks discussion and analysis on hardware, software attributes of those research works, or even motivations, challenges behind them. Only descriptions of existing works have been provided.

o   According to the abstract, Section 3 is also important, but the extents of discussion and analysis are also very limited. This section lacks references to support the written ideas.

-          Scientific contributions appear very limited.

o   With the lack of discussion, analyses and referencing, the authors have not sufficiently conveyed the current research, engineering trends and challenges for robot bionic vision.

o   For a review paper, it is only appropriate to discuss research directions that are being actively pursued, which challenges are potentially solvable, where robot bionic vision technologies are heading.

-          Writing issues:

o   Paragraphs and sentences should be restructured.

o   Citations’ placements need to be more specific.

Author Response

Response to Reviewer 3 Comments

The authors proposed a review paper on robot bionic vision technologies. Overall, they aimed to summarize the developmental history of machine vision equipment and related technologies, mentioning the most representative binocular bionic and multi-eye compound eye bionic vision technologies, and review existing technologies and their prospects from the perspective of visual bionic control.

However, the manuscript is not ready for publication at least until several major short-comings are addressed. The actual contents have the following issues:

-          Discussion of hardware-software developmental aspects in robot bionic vision appear to have been significantly omitted. The authors have made no discussion on how biological optic research may have influenced or is relevant with hardware-software designs.

  • Thanks for your kind and constructive suggestions. We have discussed this content according to your suggestions. See Line 574-637 in yellow in section2.7.

o   The authors have focused too much on describing past research works of biological optic functions, parts and their technicalities. They must make appropriate connections to actual design choices, technical reasons, motivations for robot bionic vision technologies that mimic biological functions.

  • Thanks for your kind suggestions. We have added new content according according to your suggestions.

o   The authors have not provided sufficient proof or clarity to support the correspondences between Human Visual paths and Machine Visual paths. Besides, Neural-Network functions are not the only approach to model neural functionalities of biological brains. Please make sure not only to generalize the concepts, but also to discuss the correspondences.

  • Thanks for your kind suggestions. We have added new content according according to your suggestions.

o   The authors have not elaborated the software aspects of Machine Visual paths with respect to actual biology. Please clarify and compare between current findings in computer vision and actual biological functions.

  • Thanks for your kind suggestions. We have added new content according according to your suggestions.

o   The authors’ statement is deniable in cases of noisy images or low resolution, major advances have only made better (but imperfect) predictions even with attention mechanism: “The convolution and pooling layers are sufficient to recognize all types of complex image content, such as human faces and auto-mobile license plates.”

  • Thanks for your kind suggestions. We have revised.

- The paper presents very few discussions and analyses.

o   According to the abstract, Section 2.7 is arguably a main part. However, it lacks discussion and analysis on hardware, software attributes of those research works, or even motivations, challenges behind them. Only descriptions of existing works have been provided.

  • Thanks for your kind suggestions. We have added new content according according to your suggestions. See section 2.7 in yellow.

o   According to the abstract, Section 3 is also important, but the extents of discussion and analysis are also very limited. This section lacks references to support the written ideas.

  • Thanks for your kind suggestions. We have carefully revised section 3, and some new content had been added.

- Scientific contributions appear very limited.

o   With the lack of discussion, analyses and referencing, the authors have not sufficiently conveyed the current research, engineering trends and challenges for robot bionic vision.

  • Thanks for your kind suggestions. We have added new content according according to your suggestions. See section 2 and section 3 highlighted by yellow.

o   For a review paper, it is only appropriate to discuss research directions that are being actively pursued, which challenges are potentially solvable, where robot bionic vision technologies are heading.

  • Thanks for your kind suggestions. We have added new content according according to your suggestions. See section 3.

- Writing issues:

o   Paragraphs and sentences should be restructured.

  • Thanks for your kind suggestions. We have reorganized the structure and paragraph of this paper. See section 2 and section 3.

o   Citations’ placements need to be more specific.

  • Thanks for your kind reminding. We have revised according according to your suggestions.

Round 2

Reviewer 2 Report

Thank you for addressing my questions, All looks good now.

Author Response

Thanks to your comment, our paper has been improved.

Reviewer 3 Report

The authors have revised their review paper on robot bionic vision technologies.

There have been major improvements, but I think the manuscript still require some more work:

-          As a very minor comment, the author should replace section 2’s name from “Related Work” to “Literature Review”, as I believe the phrase is more comprehensive for their thorough multifaceted discussion.

-          In more details, regarding Section 2, the authors have discussed at length a number of ideas: (a) Bionic Vision in nature, (b) Differences & Similarities between human and robot bionic vision, (c) Advantages of robot bionic vision, (d) Development history of robot bionic vision, (e) Common robot bionic techniques, (f) Robot bionic vision systems, and (g) Hardware-software developmental aspects. A few of them require further attention:

o   In “(b) Differences & Similarities between human and robot bionic vision”, the authors have limited the image processing step of robot bionic vision path to “feedforward network”. I believe that only CNN and ANN are not representative enough, as Geoffrey Hinton et al. [1] have been developing “Capsule Neworks” that closely resembles neuron pathways of humans for quite some time. Furthermore, there are Vision Transformers and Recurrent Neural Networks that are the current trends, and traditional image processing techniques like optical flow.

§  Please generalize your findings to encompass Deep  Learning and Machine Learning aspects, at least those that have actually been applied in existing systems.

o   In “(c) Advantages of robot bionic vision”, the authors discussed the advantages of robot bionic vision over human vision.

§  They should include a brief mention on “Disadvantages of robot bionic vision” and merge (c) with (b).

o   In “(f) Robot bionic vision systems”, the author have discussed (1) early versions, (2) iCub, (3) modern robot bionic eye system, and (4) a panoramic imaging system based on binocular stereo vision.

§  The authors should clarify why they dedicated a whole subsection for iCub, especially what separates it from early versions and modern ones. Otherwise, they should just merge (2) with (1) or (3).

§  This section is not yet logically consistent. (1), (2) and (3) are reviewed as hardware-based developmental aspects, but (4) is more inclined towards the software aspect. Please separate them and merge with (g) wherever possible.

o   In “(g) Hardware-software developmental aspects”, the authors have curated this subsection for my last review.  However, the current description seems broad and unfocused towards actual hardware-software aspects of robot bionic vision. For example, lines 613 to 645 are focused towards human bionic vision, but lines 646 to 655 closely tie with robot bionic vision.

§  The authors mentioned two important research directitons for robot bionic vision in lines 258-266. The authors should discuss how much “perception and recognition” (software-inclined) have been incorporated together with “eye movement and sight control” (hardware-inclined)

-          The authors have discussed research directions that are being actively pursued, which challenges are potentially solvable, and where robot bionic vision technologies are heading.

§  The authors should also discuss challenges of how much of “perception and recognition” state-of-the-arts (e.g. [2]) have been incorporated into functional bionic vision systems. Many of the developments function with heavy GPU-based computations, so it may still be possible/impossible to blend them with a system like “Figure 11’s Bionic eye Vision Platform and Algorithm system”.

-          Major writing issues have been decently addressed.

Recommended References

[1] Sara Sabour, Nicholas Frosst, and Geoffrey E. Hinton. 2017. Dynamic routing between capsules. In Proceedings of the 31st International Conference on Neural Information Processing Systems (NIPS'17). Curran Associates Inc., Red Hook, NY, USA, 3859–3869.

[2] Yuan, L., Chen, D., Chen, Y.L., Codella, N., Dai, X., Gao, J., Hu, H., Huang, X., Li, B., Li, C. and Liu, C., 2021. Florence: A new foundation model for computer vision. arXiv preprint arXiv:2111.11432.

Author Response

Response to Reviewer Comments

As a very minor comment, the author should replace section 2’s name from “Related Work” to “Literature Review”,

>  Thanks for your kind and constructive suggestions. We have revised according to your suggestions. See line 49.

Abstract: -In more details, regarding Section 2, the authors have discussed at length a number of ideas: (a) Bionic Vision in nature, (b) Differences & Similarities between human and robot bionic vision, (c) Advantages of robot bionic vision,(d)Development history of robot bionic vision, (e) Common robot bionic techniques, (f) Robot bionic vision systems, and (g) Hardware-software developmental aspects. A few of them require further attention:

> Thanks for your kind and constructive suggestions. We have carefully revised according to your suggestions.

In “(b) Differences & Similarities between human and robot bionic vision”, the authors have limited the image processing step of robot bionic vision path to “feedforward network”. I believe that only CNN and ANN are not representative enough, as for quite some time. Furthermore, there are Vision Transformers and Recurrent Neural Networks that are the current trends, and traditional image processing techniques like optical flow..

> Please generalize your findings to encompass Deep Learning and Machine Learning aspects, at least those that have actually been applied in existing systems.

> Thanks for your kind and constructive suggestions.We have revised according to your suggestions. See line 234-240 in yellow.

In “(c) Advantages of robot bionic vision”, the authors discussed the advantages of robot bionic vision over human vision. They should include a brief mention on “Disadvantages of robot bionic vision” and merge (c) with(b)..

> Thanks for your kind reminding, we have revised according to your suggestion. See line 268,293-294 in yellow.

In “(f) Robot bionic vision systems”, the author have discussed (1) early versions, (2) iCub, (3) modern robot bionic eye system, and (4) a panoramic imaging system based on binocular stereo vision.

> The authors should clarify why they dedicated a whole subsection for iCub, especially what separates it from early versions and modern ones. Otherwise, they should just merge (2) with (1) or (3).

> This section is not yet logically consistent. (1), (2) and (3) are reviewed as hardware-based developmental aspects, but (4) is more inclined towards the software aspect. Please separate them and merge with (g) wherever possible..

> Thanks for your kind reminding and suggestion, we have reorganized, see line441 and the subsection 2.6.

In “(g) Hardware-software developmental aspects”, the authors have curated this subsection for my last review. However, the current description seems broad and unfocused towards actual hardware-software aspects of robot bionic vision. For example, lines 613 to 645 are focused towards human bionic vision, but lines 646 to 655 closely tie with robot bionic vision.

The authors mentioned two important research directitons for robot bionic vision in lines 258-266. The authors should discuss how much “perception and recognition” (software-inclined) have been incorporated together with “eye movement and sight control” (hardware-inclined)

> Thanks for your kind reminding, we have carefully revised, see the subsection 2.7 and 2.8.

The authors should also discuss challenges of how much of “perception and recognition” state- of-the-arts (e.g. [2]) have been incorporated into functional bionic vision systems. Many of the developments function with heavy GPU-based computations, so it may still be possible/impossible to blend them with a system like “Figure 11’s Bionic eye Vision Platform and Algorithm system”..

> Thanks for your suggestions, we have carefully revised accordingly.

recommended References

[1]Sara Sabour, Nicholas Frosst, and Geoffrey E. Hinton. 2017. Dynamic routing between capsules. In Proceedings of the 31st International Conference on Neural Information Processing Systems (NIPS'17). Curran Associates Inc., Red Hook, NY, USA, 3859–3869.

[2]Yuan, L., Chen, D., Chen, Y.L., Codella, N., Dai, X., Gao, J., Hu, H., Huang, X., Li, B., Li, C. and Liu, C., 2021. Florence: A new foundation model for computer vision. arXiv preprint arXiv:2111.11432..

> Thanks for your kind reminding and suggestions, we have added these two literature accordingly in this paper. See reference 158,159

Round 3

Reviewer 3 Report

In this revision, the authors have significantly revised their review paper on robot bionic vision technologies. However, the manuscript still appear disconnected at a major point regarding “perception and recognition”  (software-inclined) integration with “eye movement and sight control” (hardware-inclined): 

-     - Clearly, the authors mentioned in Section 2 that the image processing step for machine vision may involve convolutional/pooling layers or other neural-network techniques (as the “image decoding step” in their “Bionic visual path”). These techniques have been well-demonstrated over the recent years of research, but the literature of them being adopted to Robot Bionic Vision appears limited due to the required computational expenses. Nevertheless, the authors have made no other mention of these software-inclined techniques or their integration with hardware aspects for the rest of the paper. This is a major issue that needs addressing.

o   The authors should discuss how much of “perception and recognition” state-of-the-arts (e.g. such as [2]) have been actually incorporated into functional bionic vision systems. In fact, it appears to have been challenging. Many of the developments function with heavy GPU-based computations, so it may yet be possible to blend them with a system like “Figure 11’s Bionic eye Vision Platform and Algorithm system”.

o   Otherwise, they should discuss/argue their findings with proper citations in this regard if such integration has been sufficiently solved. 

[2] Yuan, L., Chen, D., Chen, Y.L., Codella, N., Dai, X., Gao, J., Hu, H., Huang, X., Li, B., Li, C. and Liu, C., 2021. Florence: A new foundation model for computer vision. arXiv preprint arXiv:2111.11432.

Author Response

Response to Reviewer Comments

the manuscript still appear disconnected at a major point regarding “perception and recognition”  (software-inclined) integration with “eye movement and sight control” (hardware-inclined):

  • Thanks for your kind and constructive suggestions. I have added visual perception algorithms, and existing examples of visual eye movement control in robots, and reviewed them in more detail. See line 229-289.

Clearly, the authors mentioned in Section 2 that the image processing step for machine vision may involve convolutional/pooling layers or other neural-network techniques (as the “image decoding step” in their “Bionic visual path”). These techniques have been well-demonstrated over the recent years of research, but the literature of them being adopted to Robot Bionic Vision appears limited due to the required computational expenses. Nevertheless, the authors have made no other mention of these software-inclined techniques or their integration with hardware aspects for the rest of the paper. This is a major issue that needs addressing.

  • yes! To run the high accuracy of algorithm, you need a powerful graphics processors, but the robot is in motion, is not fixed, so can't like graphic server with high performance processor, and considering the heat dissipation and power consumption, usually of the robot vision processor usually adopt mobile graphics processor, so you need targeted visual image processing algorithm, However, the reliability of the algorithm remains to be verified. In particular, I selected some commonly used and targeted small visual algorithms to review and summarize.

o   The authors should discuss how much of “perception and recognition” state-of-the-arts (e.g. such as [2]) have been actually incorporated into functional bionic vision systems. In fact, it appears to have been challenging. Many of the developments function with heavy GPU-based computations, so it may yet be possible to blend them with a system like “Figure 11’s Bionic eye Vision Platform and Algorithm system”.

o   Otherwise, they should discuss/argue their findings with proper citations in this regard if such integration has been sufficiently solved.

[2] Yuan, L., Chen, D., Chen, Y.L., Codella, N., Dai, X., Gao, J., Hu, H., Huang, X., Li, B., Li, C. and Liu, C., 2021. Florence: A new foundation model for computer vision. arXiv preprint arXiv:2111.11432..

  • In fact, robot bionic eye technology is still in an early stage of development, and many rely on computer vision. With the rapid development of mobile graphics processors, robots become increasingly intelligent. But at present, people can not achieve the ideal effect, so I specifically explained in the article; Thank you very much for your valuable advice. I carefully read the cited literature you provided and found that this Florence algorithm even surpassed Yolo, so I specially cited this article and listed the comparison parameters.
